# Communication Efficient Federated Learning over Wireless Channels using Robust Count Sketches

## Abstract

Large-scale federated learning (FL) over wireless multiple access channels (MACs) has emerged as a crucial learning paradigm with a wide range of applications. However, its widespread adoption is hindered by several major challenges, including limited bandwidth shared by many edge devices, noisy and erroneous wireless communications, and heterogeneous datasets with different distributions across edge devices. To overcome these fundamental challenges, we propose Federated Proximal Sketching (FPS), a novel federated learning algorithm specifically designed for noisy and bandlimited wireless environments. FPS uses a count sketch data structure to address the bandwidth bottleneck and enable efficient compression while maintaining accurate estimation of significant coordinates. Moreover, FPS is designed to explicitly address the bias induced by communications over noisy wireless channels. We establish the convergence of the FPS algorithm under mild technical conditions. It is worth noting that FPS is able to handle high levels of data heterogeneity across edge devices. We complement the proposed theoretical framework with numerical experiments that demonstrate the stability, accuracy, and efficiency of FPS in comparison to state-of-the-art methods on both synthetic and real-world datasets. Overall, our results show that FPS is a promising solution to tackling the above challenges of FL over wireless MACs.

## 1 Introduction

In recent years, federated learning has emerged as an important paradigm for training high-dimensional machine learning models when the training data is distributed across several edge devices. However, when training is carried out over wireless channels in a federated setting, a number of challenges arise, including bandwidth limitations, unreliability and noise in communication channels, and statistical heterogeneity (non-identical distribution) in data across edge devices Kairouz et al. (2021). In what follows, we elaborate on three key challenges. Firstly, with the size of real world datasets and the machine learning model parameters scaling to the order of millions, communicating model parameters from edge devices to the server and back can become a major bottleneck in model training if not handled efficiently. Needless to say, the transmission of model parameters to the central server over wireless channels is noisy and unreliable in nature. In practice, channel noise is inevitable during the training process and will induce bias in learning the global model parameters. Furthermore, the data collected and stored across edge devices is heterogeneous, which adds an extra layer of complexity due to diversity in local gradient updates. If statistical heterogeneity across edge devices is not handled properly, it can significantly extend the training time and cause the global model to diverge, resulting in poor and unstable performance. Therefore, it is of significant importance to design FL algorithms that are resilient to heterogeneous data distributions and reduce communication costs. While there exists siloed efforts investigating the impacts of the above fundamental challenges separately, we devise a holistic approach - Federated Proximal Sketching (FPS) - that tackles these challenges in an integrated manner.

To address the first key challenge of communication bottleneck, we propose the use of count sketch (CS) Charikar et al. (2002) as an efficient compression operator for model parameters, as illustrated in Figure 1. The CS data structure is not only easy to implement but also comes with strong theoretical guarantees on the recovery of significant coordinates or heavy hitters. The CS data structure also enables

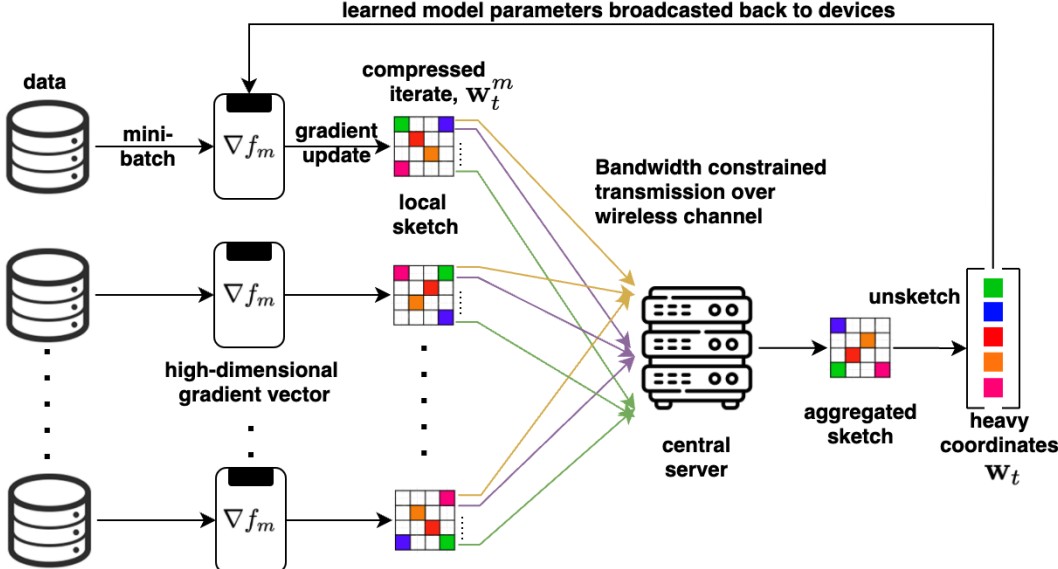

Figure 1: Illustration of Federated Proximal Sketching (FPS) over wireless multi-access channel (MAC).

us to apply the gradient updates easily such that at every time instant we preserve information of the most important model parameters. With such a compressed representation of the model parameters, over-the-air computing is then employed to aggregate local information transmitted by each device. Specifically, over-the-air Abari et al. (2016); Goldenbaum & Stanczak (2013) takes advantage of the superposition property of wireless multiple access channels, thereby scaling signal-to-noise ratio (SNR) well with an increasing number of edge devices.

To tackle the challenges due to noisy and erroneous wireless environments and data heterogeneity, we employ the proximal gradient method to 'restructure' the design of the loss function, by adding a regularization term. The regularization term is carefully selected such that it results in the following two benefits: 1) to reduce the effect of noise and 2) to keep the learned model parameters from diverging in presence of data heterogeneity. Moreover, we demonstrate empirically that this modification to our loss function helps us reduce the number of communication rounds to the central server while still maintaining high accuracy.

The main contributions of this paper can be summarized as follows:

- **Federated Proximal Sketching.** We propose Federated Proximal Sketching (FPS), a novel and robust count-sketch based algorithm for federated learning in noisy wireless environments. FPS is designed to be highly communication-efficient and can effectively handle high-level data heterogeneity across edge devices. Additionally, FPS is capable of mitigating the impact of bias induced by noisy wireless channels.

- **Impact of Gradient Estimation Errors.** Because the communications of gradient updates over noisy wireless channels may result in bias, we consider a general biased stochastic gradient structure and quantify the impact of gradient estimation errors (including bias)). We show that in the presence of biased gradient updates, the FPS algorithm converges with high probability to a neighborhood of desired global minimum, where the size of the neighborhood hinges upon the bias induced, under mild assumptions. Note that the biased stochastic gradient structure here is more general than the existing line of works on FL Stich et al. (2018); Ivkin et al. (2019); Karimireddy et al. (2020), which do not address the bias in the stochastic gradients, a key aspect in a large number of practical problems.

- **Statistical Heterogeneity:** We theoretically investigate the impact of varying degrees of statistical heterogeneity in data distributed across devices on the convergence. Our study is motivated by Li et al. (2020) to tackle data heterogeneity and extends it to a bandlimited noisy wireless channel setting. A key insight we have derived in developing FPS is that an interplay exists between the degree of data heterogeneity, rate of convergence, and choice of learning rate.

- **Experimental Studies:** We complement our theoretical studies with numerical experiments on both synthetic and real-world datasets. Our experimental results unequivocally demonstrate that FPS exhibits robust performance under noisy and bandlimited channel conditions. To evaluate the performance of our algorithm under varying degrees of class imbalance across edge devices, we have investigated different data partitioning strategies. Our results show that, in practice, our algorithm achieves high compression rates on large-scale real-world datasets without significant loss in accuracy under different data distribution strategies. In fact, in some cases, we have observed an improved accuracy of more than $10\text{-}40\%$ over other competing FL algorithms in highly heterogeneous settings.

## 2    Related Work

Our work looks at federated learning under three key challenges: (1) limited bandwidth across edge devices; (2) noisy wireless MACs; and (3) heterogeneous data distribution across devices. In what follows, we elaborate on different works which have addressed these three challenges until now.

### 2.1    Communication efficient federated learning

Over the years communication-efficient stochastic gradient descent (SGD) techniques have been developed which reduce the cost of transmission using various gradient compression techniques like quantization Bernstein et al. (2018); Wu et al. (2018); Alistarh et al. (2017), sparsification Stich et al. (2018); Aji & Heafield (2017). Different sparsification methods like $\text{top}-k$ (in absolute value) and $\text{random}-k$ have been shown to converge in theory and empirical studies. However, such sparsification methods rely on the ability to store error accumulated by the compression scheme locally and re-introduce it in the next iteration to facilitate convergence Karimireddy et al. (2019). A major limitation of $\text{top}-k$ sparsification is the additional rounds of communication between local edge devices to arrive at a consensus of global $\text{top}-k$ (heavy hitters) coordinates at each iteration. In a bandlimited setting where the number of edge of devices is large, this scheme is practically infeasible.

Our work focuses on extending the current research on applying sketching as a compression scheme in federated learning. In Ivkin et al. (2019), a communication efficient SGD algorithm was proposed which uses sketches to compress the high-dimensional gradient vectors across each of the edge devices using a count sketch data structure. However, their algorithm involves a second round of communication between the edge devices and central server to aid the estimation of $\text{top}-k$ coordinates. In practice, the second round of communication is not always feasible due to latency issues and bandwidth limitations. In Rothchild et al. (2020) as well, the authors proposed an algorithm - FetchSGD, which used sketching as a compression operator and achieved convergence without the additional rounds of communication. However, an additional error accumulation count sketch data structure has to be maintained at the central server to facilitate convergence. In addition, the work claims that FetchSGD performs well when data is distributed in a non-IID manner across edge devices but fails to provide any algorithmic details on how it deals with heterogeneous data distribution. It also lacks a detailed theoretical and practical analysis of the algorithm in different data heterogeneity scenarios which we provide in our study. While the work in Ivkin et al. (2019); Rothchild et al. (2020) aim to use sketches as a mere compression operator, we are motivated by the work in Aghazadeh et al. (2018b) which utilizes the count sketch data structure to perform SGD recursively and thus, eliminating the need to have any additional CS data structures for error accumulation. In short, we add the gradient updates in the CS data structure at every time step where they are aggregated with all the past gradient updates, leaving us with an compressed representation of model parameters. The original work in Aghazadeh et al. (2018b) was implemented for a single device (see Appendix B for more details) and we extend this to a federated learning in a band-limited noisy wireless channel setting.

## 2.2 Federated learning over wireless channels

In the previous section, the focus was on communication efficient FL under a noiseless channel setting. In practice, the transmission of gradient vectors over wireless channels to the central server is noisy and erroneous. As a consequence of transmission over noisy channels there is bias induced in gradient update vectors transmitted. The authors of Ang et al. (2020) consider regularization based optimization of the loss function to mitigate the bias induced by wireless communications. The motivation for regularization based method stems from the works of Graves (2011); Goodfellow et al. (2016), where training with noise was approximated via regularization to enhance the robustness of neural networks. There are many regularizers that one can choose from, however, there is no one regularizer which is better than the rest to tackle noise. In other words, we need to choose a regularization term specific to our problem. Due to its simplicity and ease of implementation, we use $\ell_2$-regularization.

To provide a holistic view of other related work of FL in wireless channel setting, an additional practical challenge considered is mitigating the effect of bias induced due to channel noise under limited power budget. Under such constraints, the authors in Zhang et al. (2021); Amiri & Gündüz (2020) developed an adaptive power allocation strategy based on channel state information and magnitude of gradient vector coordinates to reduce the impact of communication error on convergence results (also see Yang et al. (2020); Zhu et al. (2019)). While the above works considered only uplink channel noise, more recently, in Wei & Shen (2021), the authors analyzed the convergence of the well known FedAvg algorithm McMahan et al. (2016) under both noise in uplink and downlink transmission channels. While in this paper, we do not consider power constraints and any knowledge of channel state information, our work can be easily extended to a power constraint setting.

## 2.3 Statistical heterogeneity across edge devices

One of the fundamental challenges in federated learning as stated in Section 1 is statistical heterogeneity in data across edge devices. Recent years have witnessed the development of algorithms, such as FedProx Li et al. (2020), FedNova Wang et al. (2020b) and SCAFFOLD Karimireddy et al. (2020) to handle statistical heterogeneity. The algorithms listed above aim to reduce the drift of local iterates at each client from the global iterate maintained at the central server. The theoretical analysis of the convergence of the above-mentioned algorithms has also been well-studied under various assumptions that captures the dissimilarity in gradient computation across edge devices due to non-IID data distribution Kairouz et al. (2021). We use the bounded gradient dissimilarity assumption used in Li et al. (2020) and it has been shown to be analogous to other commonly used dissimilarity assumptions like the bounded inter-client variance Li et al. (2021b). However, these algorithms have not been studied in a band-limited and noisy wireless communication channel setting. The strategy used in FedProx is of particular interest to us, as it tackles the issue of statistical heterogeneity by appending a proximal term to the loss function. Building up on this, in later sections we show that the proximal term in our algorithm will serve two purposes; firstly, to reduce the effect of channel noise and secondly, to aid convergence in presence of statistical heterogeneity.

On a more practical side, recently a survey Li et al. (2021a) carried out an extensive experimental study on the above state-of-the-art algorithms over different data partitioning strategies and datasets. A particular kind of data partitioning strategy which is of interest to us is the label distribution skewness. A motivating example can be that some hospitals are specialized in certain kind of diseases and have data specific to it. An extreme case of label distribution skewness is where edge devices have access to only a few classes of labels Yu et al. (2020). Other notion of label skewness which is referred to as class imbalance in modern machine learning literature was studied in Wang et al. (2020b); Wang et al. (2020a); Yurochkin et al. (2019). We simulate different degrees of statistical heterogeneity by varying the amount of class imbalance present at each edge device. We believe that our work uniquely sits at the intersection of analyzing and tackling the three key FL challenges specified above.

# 3 Preliminaries

## 3.1 Federated Learning over Wireless MACs

We consider a federated learning setup where there are $M$ edge devices and a central server. Only a fraction of the dataset $\mathcal{D}$ is available across each of the edge devices such that: $\mathcal{D} = \bigcup_{m=1}^{M} \mathcal{D}_m$. The loss function at an edge device $m$ is defined as: $\ell_m(\mathbf{w}; \mathbf{x}_j, y_i)$, for a data sample $(\mathbf{x}_j, y_j) \in \mathcal{D}_m$. For a mini-batch sampled at each device $m$, the loss function is defined as:

$$f_m(\mathbf{w}; \xi^m) \triangleq \frac{\ell_m(\mathbf{w}; \xi^m)}{|\xi^m|}, \tag{1}$$

where, $|\cdot|$ represents cardinality of a set. The objective is to minimize the global loss function given by:

$$\min_{\mathbf{w} \in \mathbb{R}^d} f(\mathbf{w}) := \frac{1}{M} \sum_{m=1}^{M} \mathbb{E}_{\xi^m} \left[ f_m(\mathbf{w}; \xi^m) \right]. \tag{2}$$

Here, the expectation is taken with respect to the random process that samples mini-batches at each edge device. Such an optimization is performed iteratively to converge to the optimal model parameter vector $\mathbf{w}^*$. At each edge device $m$ and time step $t$, the stochastic gradient is computed using the sampled mini-batch $\xi_t^m$ and represented as $\mathbf{g}_t^m(\mathbf{w}_t) := \nabla f_m(\mathbf{w}_t; \xi_t^m)$. Without loss of generality, we simplify the notation of $\mathbf{g}_t^m(\mathbf{w}_t)$ to $\mathbf{g}_t^m$. The gradients are now transmitted over noisy multiple subcarriers via over-the-air protocol. We define the aggregated received gradient vector as: $\mathbf{g}_t := \frac{1}{M} \sum_{m=1}^{M} \mathbf{g}_t^m + \mathbf{n}_t$. Here, $\mathbf{n}_t \in \mathbb{R}^d$ is the channel noise. The gradient descent update rule is carried out at the central server as:

$$\mathbf{w}_{t+1} = \mathbf{w}_t - \gamma \, \mathbf{g}_t, \tag{3}$$

where, $\gamma$ is the fixed learning rate and $\mathbf{w}_{t+1}$ is model parameter vector. The updated iterate $\mathbf{w}_{t+1}$ is broadcasted back to all the edge devices. The computation of local stochastic gradients, transmission to the central server and broadcast of the updated iterates is performed recursively until we reach a small neighborhood around the global minimum $\mathbf{w}^*$. In general, transmission over wireless channels is noisy and the number of subcarriers are limited due to bandwidth constraints. As a consequence, the received gradient vector $\mathbf{g}_t$ is biased. Next, we elaborate on the count sketch compression operator and its recovery guarantees.

## 3.2 Count Sketch

A count sketch $S$ is a randomized data structure that keeps a matrix of buckets (or bins): $w \times b \sim \mathcal{O}(\log d)$, where $b$ and $w$ are chosen by the user to achieve certain accuracy guarantees. The count sketch algorithm uses $w$ random hash functions $h_j$ for $j \in [w]$ to map the vector's coordinates to buckets (or bins) $b$, $h_j : \{1, 2, \ldots, d\} \to \{1, 2 \ldots b\}$. In addition, the algorithm uses $w$ random sign functions $s_j$ for $j \in [w]$ as well that maps the coordinates of the vector randomly to $\{+1, -1\}$, $s_j : \{1, 2, \ldots d\} \to \{+1, -1\}$.

Consider a high-dimensional vector $\mathbf{g} \in \mathbb{R}^d$, then, the count sketch data structure $S$ sketches the $i^{th}$ coordinate of the vector $\mathbf{g}$ denoted as $\mathbf{g}(i)$, into the cell $S(j, h_j(i))$ by incrementing the value of the cell by $s_j(i) \, \mathbf{g}(i)$. This is performed for every $j \in [w]$ and every coordinate $i \in [d]$. Originally, as count sketch was implemented in streaming data setting, for $T$ updates to the vector $\mathbf{g}$, the count sketch data structure requires only $\mathcal{O}\left( \left( k + \frac{||\mathbf{g}_{tail}||^2}{\varepsilon^2 \, \mathbf{g}(k)} \right) \log d \, T \right)$ memory to provide unbiased estimate of the top-$k$ or heavy hitter (HH) coordinates such that the following holds with high probability:

$$|\hat{\mathbf{g}}(i) - \mathbf{g}(i)| \leq \varepsilon \, ||\mathbf{g}||, \ \ \forall i \in \mathrm{HH}, \tag{4}$$

where, HH is the set of indices of heavy hitter or top-$k$ coordinates. All norms denoted as $||\cdot||$ are $\ell_2$ norm in the Euclidean space, unless otherwise stated. Fundamental results on the recovery guarantees of count sketch can be found in Charikar et al. (2002).

We caution that the vector being sketched (here, $\mathbf{g}$) should not have too many heavy hitter coordinates. If all the coordinates of a vector are heavy, the CS data structure will have coordinates colliding and the resulting unsketched vector would be error-prone.

## 4 Federated Proximal Sketching

The key steps of the FPS algorithm are outlined in Algorithm 1. In the following, we elaborate on the key ideas further.

In Steps 1 and 2 of Algorithm 1, CS data structures at each of the edge devices and the central server are initialized to zero. Note that the size of the CS data structures is determined by the bandwidth available (number of subcarriers, $K$). We proceed with a fixed learning rate at each iteration. The number of local epochs/iterations $E$ to be carried out before each global aggregation step is pre-determined. The appropriate choice of the number of local epochs is heuristic, and we discuss it in detail in Appendix. E.3.

In Steps 5 and 6 of Algorithm 1, the stochastic gradient is computed with respect to the mini-batch sampled at each edge device. We form the gradient update vector as: $-\gamma \mathbf{g}_t^m(\mathbf{w}_t^m)$ and sketch it into the CS data structure $S^m$ maintained at that particular device $m$. To be more specific, sketching the gradient update vector to the CS data structure is implemented by the following mathematical operation in Step 6:

$$\begin{aligned}
(-\gamma \mathbf{g}_t^m) \to S^m(\mathbf{w}_t^m) &\triangleq S^m(\mathbf{w}_t^m - \gamma \mathbf{g}_t^m(\mathbf{w}_t^m)) \\
&= S^m(\mathbf{w}_{t+1}^m).
\end{aligned} \tag{5}$$

This is precisely the gradient update rule and implementation of this rule recursively is straightforward due to the linearity property of CS data structures. Observe that this update rule which compresses the computed gradient vector in a CS data structure is reminiscent of the MISSION algorithm in Aghazadeh et al. (2018a). It is worth noting that MISSION was initially designed to operate on a single device, whereas FPS is a distributed algorithm where many instances of the MISSION algorithm are carried out in parallel. At every iteration in FPS, all edge devices maintain an efficient representation of the learned model parameter vector.

In Steps 8,9 and 10 of Algorithm 1, based on how frequently updates are pushed to the server, the CS data structure at each of the devices is transmitted over noisy wireless MAC channels. The received sketches are then aggregated. We perform the top-$k$ coordinate extraction and obtain a $k$-sparse vector: $\mathbf{w}_{t+1}$. This is now broadcasted back to the edge devices.

Steps 5-10 of Algorithm 1 are carried out recursively until convergence. As we are dealing with statistical heterogeneity across devices, aggregating updates after performing a set number of local updates helps. In cases where statistical heterogeneity is high, this strategy of performing local updates alone has been known to diverge empirically McMahan et al. (2016). To address this, we restructure our loss function and discuss advantages of such a modification.

**Loss function design.** Our restructuring follows the work in Li et al. (2020) with an added benefit of mitigating the effects of channel noise. The new loss function at each device is then given by:

$$f(\mathbf{w}, \mathbf{w}^{gb}) = \ell(\mathbf{w}) + \frac{\mu}{2} \left|\left|\mathbf{w} - \mathbf{w}^{gb}\right|\right|^2, \tag{6}$$

where, $\ell(\mathbf{w})$ is our application specific loss function, for instance, a cross-entropy loss for binary classification task or a mean-squared error for linear regression task. We denote the iterate $\mathbf{w}^{gb}$ as the last aggregated model parameter vector that was broadcasted by the central server. Therefore, for a non-zero proximal parameter $\mu$, this new loss function provides the following benefits; 1) it controls the effect of statistical heterogeneity across devices by not letting the local updates $\mathbf{w}$ stray far away from the last global update $\mathbf{w}^{gb}$, 2) for improperly chosen number of local updates $E$, the proximal term minimizes the effect of divergence that would result as a consequence and, 3) it provides a regularization effect on the global iterates and thus we can bound the $\ell_2$ norm by some arbitrary positive constant, $||\mathbf{w}^{gb}||^2 \leq W$.

## 5 Convergence Analysis

As is standard, the loss function $f_i$ at each edge device $i$ is assumed to be $L$-smooth non-convex objective function.

---

**Algorithm 1** Federated Proximal Sketching (FPS)

---

1: **Inputs:** Number of workers: $M$, mini-batches for each worker $m \in [M]$ at each time step: $\xi_t^m$, local epochs $E$.
2: Initialize individual sketches at each worker $S^m$ with initial model parameters $\mathbf{w}_0^m$: $\mathbf{w}_0 \rightarrow S^m = S^m(\mathbf{w}_0)$
3: **for** $t = 1, 2, \ldots, T$ **do**
4:     **for** $m = 1, 2, \ldots, M$ **do**
5:         Compute stochastic gradient using mini-batch $\xi_t^m$: $\mathbf{g}_t^m(\mathbf{w}_t^m)$
6:         Sketch the gradient update vector $(-\gamma \mathbf{g}_t^m)$ at each worker: $(-\gamma \mathbf{g}_t^m) \rightarrow S^m(\mathbf{w}_t^m) = S^m(\mathbf{w}_{t+1}^m)$ and broadcast it to the central server after $E$ local iterations / epochs
7:     **end for**
8:     Receive aggregated sketches at the server: $S_t(\mathbf{w}_{t+1}) = \frac{1}{M} \sum_{m=1}^{M} S^m(\mathbf{w}_{t+1}^m) + n_t$
9:     Unsketch and extract top-k coordinates of parameter vector: $\mathbf{w}_{t+1} = \mathcal{U}_k(S_t(\mathbf{w}_{t+1}))$
10:     Broadcast $k$-sparse parameter vector to all edge devices: $\mathbf{w}_{t+1}^m = \mathbf{w}_{t+1}$
11: **end for**

---

**Assumption 1** (*Smoothness*) *A function $f : \mathbb{R}^d \rightarrow \mathbb{R}$ is $L-$smooth of for all $x, y \in \mathbb{R}^d$, it holds:*

$$|f(y) - f(x) - \langle \nabla f(x), y - x \rangle| \leq \frac{L}{2} ||y - x||^2 . \tag{7}$$

In general, the received aggregate stochastic gradient $\mathbf{g}_t$, is biased, i.e., $(\mathbb{E}[\mathbf{g}_t] \neq \nabla f(\mathbf{w}_t))$, and this can be due to biased stochastic gradient estimation, data heterogeneity across devices and noisy channel conditions Zhang et al. (2021); Amiri & Gündüz (2020). In what follows, we examine the structure of stochastic gradient vector received at the central server.

**Definition 1** *Given a sequence of iterates $\{\mathbf{w}_t\}_{t=1}^T$, for all $t \in [T]$, the structure of biased stochastic gradient estimator can be written as:*

$$\mathbf{g}_t(\mathbf{w}_t) = \nabla f(\mathbf{w}_t) + \beta_t + \zeta_t , \tag{8}$$

*where, $\beta_t$ is the biased estimation error and $\zeta_t$ is the martingale difference noise. The quantities $\beta_t$ and $\zeta_t$ are defined as:*

$$\beta_t := \mathbb{E}_t[\mathbf{g}_t(\mathbf{w}_t)] - \nabla f(\mathbf{w}_t) \tag{9}$$

$$\zeta_t := \mathbf{g}_t(\mathbf{w}_t) - \mathbb{E}_t[\mathbf{g}_t(\mathbf{w}_t)] . \tag{10}$$

Note that such a structure of stochastic gradient estimator has been studied in Zhang et al. (2008); Ajalloeian & Stich (2020b). It directly follows from the above definition of bias and martingale difference noise that $\mathbb{E}[\zeta_t] = 0$. Here, the expectation $\mathbb{E}_t[\cdot]$ is with respect to $\xi_t$ which is a realization of a random variable which represents the choice of single training sample in the case of vanilla SGD or may represent a set of sample in the case of mini-batch SGD, and the channel noise $n_t$. Furthermore, we assume that the bias and martingale noise terms satisfies the following assumptions.

**Assumption 2** (*Zero mean, $(P_n, \sigma^2)$-bounded noise*) *There exists constants $P_n, \sigma^2 \geq 0$ such that:*

$$\mathbb{E}_t \left[ || \zeta_t ||^2 \right] \leq P_n ||\nabla f(\mathbf{w}_t)||^2 + \sigma^2 . \tag{11}$$

**Assumption 3** (*$(P_b, b^2)$-bounded bias*) *There exists constants $P_b \in (0, 1)$ and $b^2 \geq 0$ such that:*

$$|| \beta_t ||^2 \leq P_b ||\nabla f(\mathbf{w}_t)||^2 + b^2 . \tag{12}$$

These assumptions are significantly mild as the second moment bounds of the bias and noise terms scales with true gradient norm and constants $b^2$ and $\sigma^2$ respectively. By setting the tuple $(P_b, P_n, b^2, \sigma^2) = \bar{0}$, we

get the special case of unbiased gradient estimators. Convergence for this special case has been well studied in literature.

Next, we turn our attention to the compressibility of gradients. Specifically, we assume that the stochastic gradients are approximately sparse. This is formalized in the following assumption Cai et al. (2022).

**Assumption 4** *The stochastic gradients follow a power law distribution and there exists a $p \in (1, \infty)$ such that $|\mathbf{g}_t(i)| = i^{-p} \, ||\mathbf{g}_t||$ .*

In the Appendix, we show that some of the real-world dataset(s) considered in this paper follow Assumption 4. As the value of $p$ increases we infer that only a small number of coordinates in the vector $\mathbf{g}$ are significant. Therefore by choosing an appropriate size of CS data structure we can ensure efficient compression and strong recovery guarantees of the significant coordinates.

Even though the loss functions across all the devices are same, as the data is distributed in a non-IID manner, due to random sampling of mini-batches across devices there will be dissimilarities in computation of loss functions and their respective gradient estimators. To this end, we define a measure of dissimilarity between gradient estimators across edge devices similar to Li et al. (2020) as follows.

**Definition 2** *(B-local dissimilarity). The local functions $f_m$ are $B-$locally dissimilar at $\mathbf{w}$ if $||\mathbb{E}_{\xi_m}[\nabla f_m(\mathbf{w}; \xi_m)]||^2 \leq ||\nabla f(\mathbf{w})||^2 B^2$. We further define $B(\mathbf{w}) = \sqrt{\frac{\mathbb{E}_{\xi_m}[||\nabla f_m(\mathbf{w}; \xi_m)||^2]}{||\nabla f(\mathbf{w})||^2}}$, for $||\nabla f(\mathbf{w})|| \neq 0$.*

Further, we have the following assumption ensuring that the dissimilarity $B(\mathbf{w})$ defined in Definition 2 is uniformly bounded above.

**Assumption 5** *For some $\epsilon > 0$, there exists $B$ such that for all points $\mathbf{w} \in S_\epsilon = \{\mathbf{w} \, | \, ||\nabla f(\mathbf{w})||^2 > \epsilon\}$, $B(\mathbf{w}) \leq B$.*

If we assume the data is distributed in an IID manner, the same loss function across all devices and the ability to sample an infinitely large sample size, then, $B \to 1$. However, due to different sampling strategies, in practice, $B > 1$. A larger value of $B$ would imply higher statistical heterogeneity across devices. Other formulations of measuring dissimilarity have been studied in Khaled et al. (2019); Li et al. (2019); Wang et al. (2019).

Let us denote $H = \frac{1}{1 + 2 B^2 (P_b + P_n)}$. Note that $H \leq 1$. We now define the following quantity $\rho(\gamma)$ as:

$$\rho(\gamma) \triangleq \frac{1 - P_b (1 + 2H) E^2 B^2}{2} - 2 P_n (L + \mu) (1 + 2H) \gamma E^2 B^2 - \gamma E^2 (2 + 2P_b B^2 + P_n B^2) (1 + 2H) , \tag{13}$$

where, $P_b$, $P_n$, $L$ and $B$ are constants defined earlier; $\mu$ is the proximal parameter of our loss function and $E$ is number of local epochs carried out at each edge device before global aggregation of model parameters at the central server. Let $f(\mathbf{w}^*)$ be the global minimum value of $f$. The range of values of the fixed learning rate $\gamma$ which we consider, satisfies the following conditions: $\rho(\gamma) > 0$ and $\gamma \leq \frac{1}{4 E (L + \mu) (1 + 2 B^2 (P_b + P_n))}$. The CS data structure size we consider scales like $\mathcal{O}\left(c \, k \, \log \frac{d \, T}{\delta}\right)$. Here, $c$ is some positive scalar (c>1), $k$ denotes the number of heavy hitter coordinates we are extracting or unsketching from the CS data structure, $d$ is the ambient dimension, $T$ is the number of iterations and $\delta$ is probability of error. We bound the $\ell_2$ norm of the iterates by some arbitrary positive constant, $||\mathbf{w}||^2 \leq W$. We have the following main theorem on the iterates in the FPS algorithm.

**Theorem 1** *Under Assumptions 1, 2, 3, 4 and 5, the following result holds with probability at least $1 - \delta$:*

$$\frac{1}{T+1} \sum_{t=0}^{T} \rho(\gamma) \left\| \nabla f(\mathbf{w}_t) \right\|^2 \leq \frac{|f(\mathbf{w}_0) - f(\mathbf{w}^*)|}{\gamma(T+1)} + \left( \frac{1}{c} + \delta_1 \right) \frac{(L+\mu)^2 W^2}{2}$$

$$+ \left( \left(1 + 4 P_b B^2\right) E^2 + \frac{E}{2} + \frac{\left(1 + \left(2 + 2P_b B^2 + P_n B^2\right) E\right)}{2(L+\mu)} \right) b^2$$

$$+ \left( 2 E^2 + \left(1 + 4 P_n B^2\right) \frac{E}{2} + \frac{\left(2 \left(2 + 2P_b B^2 + P_n B^2\right) + 1\right) E}{4(L+\mu)} \right) \sigma^2, \tag{14}$$

*where, $\delta_1 < 1$.*

**Remarks.** We have a few important observations in order.

- The first term on the right hand side of equation 14 is a scaled version of the term $|f(\mathbf{w}_0) - f(\mathbf{w}^*)|$, and its effect diminishes as $T \to \infty$.

- The second term in equation 14 captures the error in unsketching of the top$-k$ coordinates of the iterates $\mathbf{w}$. It can also be viewed as the residual error after extracting top$-k$ coordinates from the CS data structure. As the CS size increases, $c$ increases and as a consequence $1/c$ is small in magnitude. The quantity $\delta_1$ is defined as the ratio $\frac{\sum_{i=k+1}^{d} i^{-2p}}{\sum_{i=1}^{d} i^{-2p}}$, $\forall k \geq 1$. Clearly, as the CS size increases, the ability to extract more coordinates increases ($k$ increases). This implies, the value of $\delta_1$ decreases. However, the rate at which $\delta_1$ decreases also depends on the dataset. The power $p$, depends on the how effectively we can represent the relation between input and output using a small subset of features. The lesser the number of features used the higher the value of $p$ and vice versa. Thus, as the bandwidth at each edge device increases, the size of CS data structure increases as well and the effect of this term can be suppressed.

- The third and fourth terms in Equation equation 14 capture the effects of bias $\beta_t$ and noise $\zeta_t$, respectively. We can observe that the algorithm will visit a neighborhood that scales by constants $b^2$ and $\sigma^2$ with high probability. Additionally, these terms scale as the values of local epochs $E$ and the degree of heterogeneity $B$ increase. To ensure convergence of the algorithm, there is no fixed value of $E$ that works for different degrees of data heterogeneity. For example, as the data distribution across edge devices becomes increasingly heterogeneous, a smaller value of $E$ can reduce the magnitude of the $B^2$ terms. This can be intuitively explained by arguing that as heterogeneity increases, choosing a higher value of local epochs will result in aggregating bias and noise due to the large dissimilarity in gradient computations across different edge devices. Therefore, as heterogeneity increases, the number of local epochs should be low to facilitate convergence. This implies more frequent communication with the central server. If bandwidth is a concern, this issue can be slightly alleviated by increasing the proximal parameter $\mu$, thereby reducing the impact of high statistical heterogeneity.

- Another aspect of our result arises from analyzing Equation equation 13, which provides a bound on the learning rate for facilitating convergence. Upon observation, we aim for the quantity defined in Equation equation 13 to be greater than zero: $\rho(\gamma) > 0$. It is noteworthy that if the dissimilarity $B$ is large, the impact of the third and fourth terms in this equation can be mitigated by selecting a very small learning rate and reducing the number of local epochs. This intuitive approach makes sense because as the dissimilarity measure increases, the probability of local models diverging from the global minimum also increases. Hence, a smaller learning rate and fewer local epochs need to be chosen to stabilize the algorithm and ensure $\rho(\gamma) > 0$. However, the second term in Equation equation 13 does not exhibit the same tunability of parameters to address larger dissimilarities. $P_b$ is a constant determined by our local optimization solver and channel noise. Therefore, if the product $P_b B^2 E^2 > 1$, the algorithm cannot converge. In other words, there is a limit to which our specified algorithm can handle statistical heterogeneity.

# 6 Experimental Studies

We conduct several experiments on synthetic and real-world datasets, with different model and environmental parameters. Under a bandlimited and noisy wireless channel setting, we simulate the performance of our proposed algorithm - FPS, and other competing bandlimited algorithms like FetchSGD, Rothchild et al. (2020) and bandlimited coordinate descent (BLCD), Zhang et al. (2021). For the count sketch based algorithms like FetchSGD and FPS, the number of subcarriers or channels will dictate the CS data structure size. In case of BLCD random sparsification is as a compression operator, therefore, the number of subcarriers will decide the number of coordinates of the gradient vector that will be selected at random for transmission to the central server. The number of edge devices $M$ for all our experiments is chosen to be 10. The channel noise over each subcarrier follows a zero mean normal distribution, $\mathcal{N}(0, 1)$. For FetchSGD and BLCD, the global aggregation to the central server is performed at every epoch / iteration as designed in the papers they were proposed in. For FPS, we perform the global aggregation after every 5 local epochs / iterations. The number of local epochs is chosen heuristically and its choice is discussed more in the Appendix E.3. We choose a learning rate of 0.01 for all our experiments. To simulate varying degrees of data heterogeneity, the following data partitioning scenarios are considered in our experiments:

**Scenario 1.** The data across all edge devices is distributed in an IID manner with equal number of samples corresponding to each class available.

The kind of non-IID distribution we consider in our work is label distribution skewness. Under the umbrella of label skewness, there are two sub-divisions of data partitioning strategy: quantity-based label imbalance and distribution-based label imbalance.

**Scenario 2.** In this case, we consider quantity-based label imbalance, where, each edge device has access to samples corresponding to fixed number of classes only. For instance, in a binary classification problem the edge devices will have access to samples corresponding to only one class.

**Scenario 3.** Here, a distribution decides the proportion of samples of each label assigned to each edge device. A natural candidate for this task is a Dirichlet distribution. A hyperparameter $\alpha$ dictates how skewed the proportion of samples of each label across the devices will be. We sample the probabilities $p_l \sim \text{Dir}_M(\alpha)$ for a particular class label $l$. The probability vector $p_l$ whose entries sum up to one, decides the proportion of samples of class $l$ across all devices. Lower values of $\alpha$ correspond to highly skewed distribution of class labels and conversely, higher values correspond to a more even distribution of samples of each class across all devices. The value of $\alpha$ we consider in this scenario is 0.1.

**Scenario 4.** In this case, the setup is the same as Scenario 3 with the value of hyperparameter for Dirichlet distribution set to $\alpha = 1$.

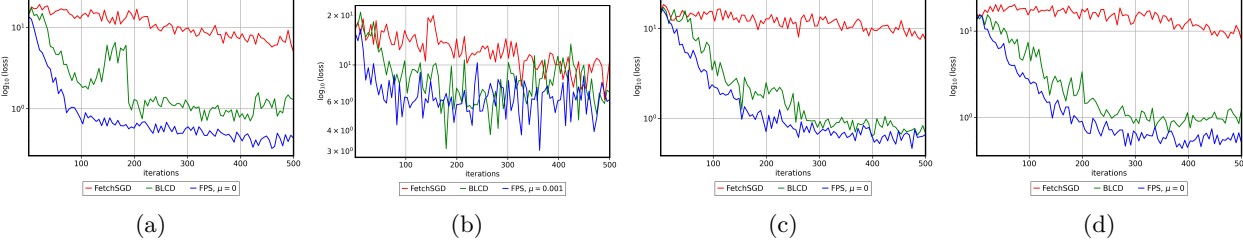

Figure 2: Plotting logarithm of test loss computed for FPS, BLCD, FetchSGD over 5 trials under noisy channel conditions with the gradients following Assumption 4 and power law degree $p = 5$. The figures correspond to different data partitioning strategies: (a) Scenario 1 (b) Scenario 2 (c) Scenario 3 (d) Scenario 4.

## 6.1 Synthetic dataset

**Data generation.** For scenario 1, consider generating observations, $\mathbf{y} = \mathbf{X}\mathbf{w} + 0.01\,\mathbf{n}$, where, $\mathbf{w} \in \mathbb{R}^d$ is the parameter vector, $\mathbf{n} \in \mathbb{R}^d$ is the additive Gaussian noise and whose each element $n_i$ distributed

according to $\mathcal{N}(0,1)$. The design matrix is denoted by $\mathbf{X} \in \mathbb{R}^{N \times d}$ where each row $\mathbf{X}_i \in \mathbb{R}^d$ is a data sample distributed according to $\mathcal{N}(\bar{\mathbf{0}}, \Sigma)$. Here, the diagonal elements of $\Sigma$ are non-zero and diminish such that $\Sigma_{ii} = i^{-p} \, \forall \, i \in [d]$.

For scenarios 2, 3 and 4, we generate equal number of observations under two different distributions, one where $\mathbf{X}_i \sim \mathcal{N}(\bar{\mathbf{0}}, \Sigma_1)$ and the other where $\mathbf{X}_i \sim \mathcal{N}(\bar{\mathbf{0}}, \Sigma_2)$. Here, $\Sigma_1 = \Sigma$ as defined in Scenario 1. We choose the other diagonal matrix $\Sigma_2$ such that the diagonal elements are $\Sigma_{ii} = j^{-p}$, here, $j$ is some random permutation of the index set $\{1, 2, \ldots, d\}$.

**Experimental setup.** The number of subcarriers allocated to each edge device are 256. For FPS, the set of values of proximal parameter we consider are: $\mu = \{0, 0.001, 0.01, 0.1\}$. The ambient dimension $d$ and power law degree $p$ are set to 10000 and 5 respectively.

We plot the average of logarithm of test loss over 10 trials under noisy bandlimited setting for FPS, FetchSGD and BLCD in Figure 2. Starting from left to right, the figures correspond to data partitioning scenarios 1, 2, 3 and 4 respectively. Across all experiments FPS achieves the lowest test loss. BLCD maintains comparable performance in Scenario 2 and a slightly weaker performance to FPS in other scenarios. Whereas FetchSGD exhibits poor performance across all scenarios. For each of the data partitioning scenarios, we mention the value of proximal parameter for which FPS performs the best in the plot legends below.

## 6.2 Real-world datasets

For our experimental study we consider two real-word datasets, both, binary classification tasks and we choose to minimize the log-loss score. The average accuracy is reported corresponding to each of the data partitioning scenarios in noisy and noise-free case over 5 trials. The best choice of proximal parameter from the set, $\mu = \{0, 0.01, 0.1, 1\}$ for each scenario is mentioned in the legend below each plot.

### 6.2.1 KDD12 - Click prediction

The KDD12 dataset is binary classification task where the model must classify if a user will accept $\{1\}$ or reject $\{0\}$ an item recommended to it. Here, the items are news, games, advertisements, products. For more details on the dataset, see Juan et al. (2016). The number of features in this dataset are $54,686,452$. The number of subcarriers allocated to each edge device is, $K = 1024$. In Figure 3, we observe that FPS

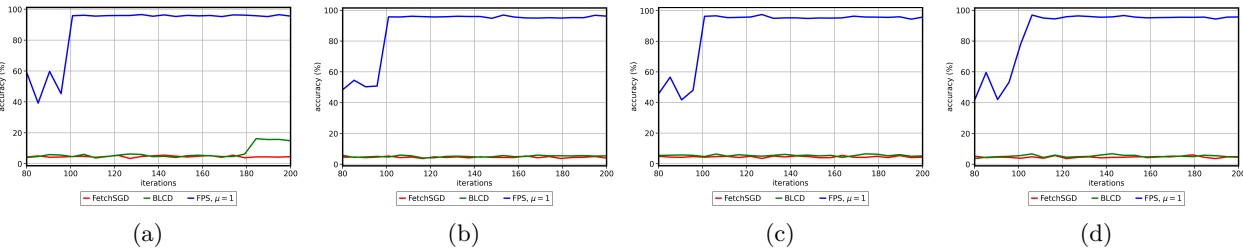

Figure 3: Plotting test accuracy for FPS, BLCD, FetchSGD on KDD12 dataset under noisy channel conditions. The figures correspond to different data partitioning strategies: (a) Scenario 1 (b) Scenario 2 (c) Scenario 3 (d) Scenario 4. We can observe that FPS converges to a global optimum quickly and outperforms other competing bandlimited algorithms by a huge margin.

performs much better compared to FetchSGD and BLCD across all data partitioning strategies and noisy channel conditions. Also, FPS converges quicker compared other competing bandlimited algorithms. In Table 1, we report the mean accuracy over 5 trials for various FL algorithms including FPS under varying degrees of statistical heterogeneity and channel noise conditions.

### 6.2.2 KDD10 - Predicting student performance

The number of features in the dataset are $20,216,830$. For more details on the dataset, see Yu et al. (2010). The number of subcarriers that are allocated to each edge device, $K = 4096$.

Similarly, in Figure 4, we observe that FPS performs much better compared to FetchSGD and BLCD across all data partitioning strategies in bandlimited noisy channel conditions. In Table 2, we report the mean accuracy over 5 trials for various FL algorithms including FPS under varying degrees of statistical heterogeneity and channel noise conditions.

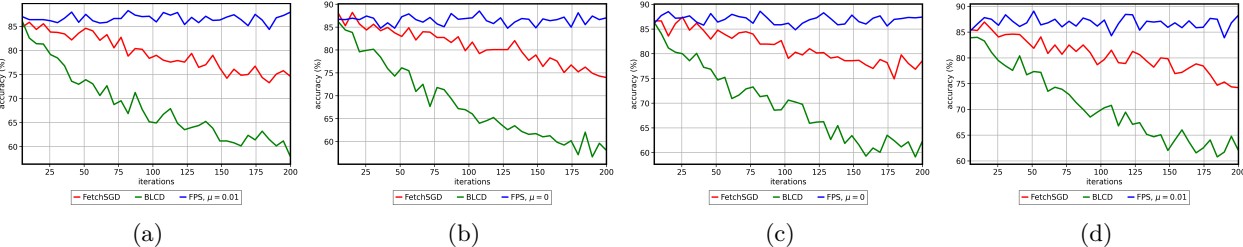

Figure 4: Plotting test accuracy for FPS, BLCD, FetchSGD on KDD10 dataset under noisy channel conditions. The figures correspond to different data partitioning strategies: (a) Scenario 1 (b) Scenario 2 (c) Scenario 3 (d) Scenario 4. We can see that FPS is stable under noisy channel conditions and consistently performs better than other competing bandlimited algorithms.

| Label skewness | Noise $\mathcal{N}(0, \sigma^2)$ | Accuracy (%) | | | | |
|---|---|---|---|---|---|---|
| | | FPS | FetchSGD | BLCD | Top-k | FedProx |
| Scenario 1 | $\sigma = 0$ | $96.44 \pm 0.81$ | $96.48 \pm 1.52$ | $8.51 \pm 2.67$ | $\mathbf{96.64 \pm 0.52}$ | $96.48 \pm 0.81$ |
| | $\sigma = 1$ | $\mathbf{96.56 \pm 1.29}$ | $5.46 \pm 1.33$ | $16.17 \pm 21.23$ | $68.82 \pm 16.66$ | $57.42 \pm 13$ |
| Scenario 2 | $\sigma = 0$ | $\mathbf{97.03 \pm 1.14}$ | $48.12 \pm 1.26$ | $5.93 \pm 1.6$ | $51.09 \pm 2.93$ | $53.20 \pm 6.99$ |
| | $\sigma = 1$ | $\mathbf{96.87 \pm 0.95}$ | $5.39 \pm 0.96$ | $5.93 \pm 1.85$ | $57.57 \pm 24.04$ | $40.93 \pm 11.12$ |
| Scenario 3 | $\sigma = 0$ | $96.64 \pm 0.52$ | $\mathbf{96.79 \pm 0.51}$ | $6.56 \pm 1.38$ | $96.64 \pm 1.22$ | $96.56 \pm 0.67$ |
| | $\sigma = 1$ | $\mathbf{97.5 \pm 0.97}$ | $5.39 \pm 1.24$ | $6.4 \pm 1.27$ | $72.57 \pm 15.3$ | $54.60 \pm 17.26$ |
| Scenario 4 | $\sigma = 0$ | $96.25 \pm 0.76$ | $96.17 \pm 1.08$ | $17.18 \pm 19.55$ | $\mathbf{96.71 \pm 0.46}$ | $96.01 \pm 1.22$ |
| | $\sigma = 1$ | $\mathbf{96.87 \pm 0.95}$ | $6.09 \pm 0.31$ | $6.79 \pm 1.93$ | $66.32 \pm 14.71$ | $46.32 \pm 10.79$ |

Table 1: Test accuracy of different distributed algorithms under varying channel conditions and statistical heterogeneity. For FPS and FedProx, we tune $\mu$ from $\{0, 0.01, 0.1, 1\}$ and report the best accuracy over KDD 12 dataset.

| Label skewness | Noise $\mathcal{N}(0, \sigma^2)$ | Accuracy (%) | | | | |
|---|---|---|---|---|---|---|
| | | FPS | FetchSGD | BLCD | Top-k | FedProx |
| Scenario 1 | $\sigma = 0$ | $88.04 \pm 1.53$ | $86.64 \pm 1.19$ | $86.79 \pm 2.45$ | $87.10 \pm 1.54$ | $\mathbf{88.12 \pm 2.35}$ |
| | $\sigma = 1$ | $\mathbf{87.96 \pm 1.36}$ | $75.78 \pm 3.84$ | $63.20 \pm 4.15$ | $55.85 \pm 6.15$ | $55.46 \pm 1.69$ |
| Scenario 2 | $\sigma = 0$ | $\mathbf{87.03 \pm 1.66}$ | $54.37 \pm 2.6$ | $72.18 \pm 4.02$ | $54.06 \pm 3.64$ | $55 \pm 1.73$ |
| | $\sigma = 1$ | $\mathbf{88.12 \pm 1.75}$ | $76.25 \pm 3.18$ | $62.03 \pm 2.81$ | $50.07 \pm 3.089$ | $56.71 \pm 3.39$ |
| Scenario 3 | $\sigma = 0$ | $\mathbf{89.68 \pm 1.75}$ | $75.54 \pm 1.68$ | $77.65 \pm 3.21$ | $78.35 \pm 3.11$ | $80.46 \pm 2.26$ |
| | $\sigma = 1$ | $\mathbf{87.42 \pm 2.05}$ | $79.76 \pm 3.40$ | $62.42 \pm 3.37$ | $52.03 \pm 6.01$ | $54.14 \pm 3.86$ |
| Scenario 4 | $\sigma = 0$ | $87.81 \pm 1.96$ | $86.25 \pm 1.44$ | $86.95 \pm 1.72$ | $88.28 \pm 1.71$ | $\mathbf{88.43 \pm 1.12}$ |
| | $\sigma = 1$ | $\mathbf{88.28 \pm 2.06}$ | $76.71 \pm 7.15$ | $64.76 \pm 2.11$ | $59.37 \pm 5.78$ | $56.32 \pm 3.6$ |

Table 2: Test accuracy of different distributed algorithms under varying channel conditions and statistical heterogeneity. For FPS and FedProx, we tune $\mu$ from $\{0, 0.01, 0.1, 1\}$ and report the best accuracy over KDD 10 dataset.

## 6.3 Discussion

For the real-world datasets considered in this paper (KDD10 and KDD12), we show that the computed stochastic gradient vector at each iteration satisfies the approximately sparse gradient assumption (Assump-

tion 4 ) in Appendix F. Specifically for KDD12, the number of significant coordinates in the gradient update vectors are extremely low compared to the ambient dimension of the dataset. In this case, algorithms like BLCD will perform poorly as the probability of randomly selecting significant coordinates when the ambient dimension is huge, is very low. This poor performance of BLCD can be be seen in Figures 3 and 4. FetchSGD on the other hand maintains an efficient representation of significant coordinates of the gradient update vectors, so one would expect it to perform well. On the contrary, as FetchSGD contains no mechanism to tackle noisy wireless channels and data heterogeneity; it's performance is poor as well. The only scenarios where FetchSGD performs comparable to our algorithm FPS, is when the data is distributed in an IID manner (scenario 1) and the degree of statistical heterogeneity is low (scenario 4), and the communication is over noise free channels (see Tables 1, 2). Accuracy plots of FetchSGD, BLCD and FPS are shown in Figures 6 and 5 in Appendix E.2.

We take our comparison a step further by evaluating FPS against FedProx Li et al. (2020) and top-k federated algorithms which are not bandlimited in nature. FedProx is one of the state-of-the-art algorithms recently published which aims to learn a global model when data heterogeneity exists across edge devices. FedProx communicates the whole gradient update vector with the central server and top-k algorithm requires extra rounds of communication between other edge devices to achieve consensus on global top-k gradient coordinates. The detailed accuracy results are given in Tables 1 and 2. When the data is extremely heterogeneous (Scenario 2), we see that FedProx and top-k do not perform well under both noisy and noise-free channel conditions. Under mild statistical heterogeneity setting like Scenario 4, we see that FedProx and top-k perform on par with our FPS algorithm in a noise-free channel setting, however, they struggle in noisy channel conditions. We hypothesize the poor performance of FedProx in noisy channel conditions due to approximately sparse gradient update vectors being corrupted by the channel noise. As the less significant coordinates are corrupted, this results in erroneous gradient updates. One can argue that this can be resolved by scaling the gradient coordinates well above the noise floor but this approach seems to be infeasible when there are power constraints imposed.

## 7 Conclusion

In this paper, we proposed Federated Proximal Sketching (FPS), a novel algorithm that learns a global model under bandlimited noisy wireless channels and when there is data heterogeneity present across edge devices. In fact, we are the first to provide both theoretical guarantees and empirical results while using sketching as a compression operator under bandlimited noisy wireless channel setting with data heterogeneity across edge devices. Theoretically, we show that the communication cost to the central server at any round is $\mathcal{O}(\log d)$ which is significantly lower than the ambient dimension $d$ when dealing with large-scale datasets. Our experiments corroborate that the count-sketch compression scheme in FPS significantly reduces the communication cost without any discernible loss in model performance.

To simulate data heterogeneity across edge devices we consider different data partitioning strategies motivated by real-world scenarios. We show that the restructuring of our loss function by appending a proximal term stabilizes and keeps FPS from diverging under varying degrees of data heterogeneity and in the presence of channel noise. Mathematically, we model the effects of data heterogeneity and bias induced due to channel noise using mild technical assumptions and provide an easy to interpret convergence result which shows an interplay between various parameters like the size of CS data structure, degree of statistical heterogeneity, magnitude of bias induced and rate of convergence.

Overall, our work adeptly tackles three of the most pressing challenges in federated learning setup: data heterogeneity across edge devices, bandlimited and noisy wireless channels, and demonstrates the robustness and efficacy of our proposed algorithm - FPS. Our experiments conducted over synthetic and large-scale real-world datasets, substantiate our theoretical guarantees and showcase the superior, stable and highly accurate performance of FPS over other state-of-the-art federated learning algorithms.

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

# A  Appendix

This appendix is organized as follows: Section B outlines a core element of our paper, the MISSION algorithm. Section C provides the main result for count sketch data structure. Section D provides detailed proofs of main theorem and lemma. Section E discusses the experimental setup and additional empirical results. Section F discusses empirical results supporting the gradient compressibility assumption (Assumption 4) made in the main paper.

# B MISSION Algorithm

The algorithm proposed in Aghazadeh et al. (2018a) is first initialized with a vector $\mathbf{w}_0$ and initialize a count sketch data structure $S$ with zero entries. At iteration $t$, mini-batch stochastic gradient is computed using mini-batch $\xi_t$ and we denoted this as $\mathbf{g}_t$. We form the the gradient update vector by multiplying it with the learning rate: $(-\gamma \mathbf{g}_t)$. We then add the non-zero entries of this computed gradient update vector to the count sketch $S$. Next, MISSION extracts top-$k$ heavy hitters from the sketch, $\mathbf{w}_{t+1}$. The process computation of stochastic gradients and adding it to the sketch is run recursively until the number of iterations desired or until convergence.

---
**Algorithm 2** MISSION
---
1: Initialize initial vector $\mathbf{w}_0$, Count Sketch $S$ and learning rate $\gamma$
2: **for** $t = 1, 2, \ldots, T$ **do**
3:     Compute stochastic gradient using mini-batch $\xi_t$: $\mathbf{g}_t(\mathbf{w}_t)$
4:     Sketch the local vector $(-\gamma \mathbf{g}_t)$ into $S(\mathbf{w}_t)$: $S(\mathbf{w}_t - \gamma \mathbf{g}_t)$
5:     Unsketch and extract parameter vector: $\mathbf{w}_{t+1} = \mathcal{U}_k(S(\mathbf{w}_{t+1}))$
6: **end for**
7: **Return:** The top-$k$ heavy-hitters of parameter vector $\mathbf{w}$ from the Count-Sketch
---

# C Count Sketch

We now state the main theorem of count sketch data structure.

**Theorem 2 (Count-sketch).** *For a vector* $\mathbf{g} \in \mathbb{R}^d$, *count sketch recovers the top-$k$ coordinates with error* $\pm\varepsilon||\mathbf{g}||_2$ *with memory* $\mathcal{O}\left(\left(k + \frac{||\mathbf{g}^{tail}||^2}{\varepsilon^2 \mathbf{g}(k)^2}\right) \log \frac{dT}{\delta}\right)$; *where* $||\mathbf{g}^{tail}||^2 = \sum_{\beta \notin top-k}(\mathbf{g}(i))^2$ *and* $\mathbf{g}(k)$ *is the $k$-th largest coordinate and this holds with probability at least* $1 - \delta$.

For a detailed proof, we refer to Charikar et al. (2002) .

# D Proofs

## D.1 Lemma

Here we state a lemma that upper bounds the residual error after unsketching top-$k$ coordinates of the iterates. This lemma follows directly from the initial recovery guarantees derived in Charikar et al. (2002). We uniformly bound the iterates above by a positive constant $W$ such that: $\mathbb{E}\left[||\mathbf{w}||^2\right] \leq W^2$. Though this might seem like a bold assumption, we empirically validate that this is true in Section F. We denote the unsketched top-$k$ coordinates of the iterate $\mathbf{w}_t$ as $\tilde{\mathbf{w}}_t$. Here, the subscript $t$ denotes the time index. Under Assumption 4 and the recovery guarantees stated in Theorem 2 we state the following lemma.

**Lemma 1** *If the Count Sketch algorithm recovers the top-$k$ coordinates with error* $\varepsilon = \frac{1}{\sqrt{c\,k}}$ *and sketch size scaling like* $\mathcal{O}\left(c\,k \log \frac{dT}{\delta}\right)$, *the following holds for any iterate* $\mathbf{w} \in \mathbb{R}^d$ *with probability at least* $1 - \delta$:

$$\mathbb{E}\left[||\mathbf{w}_t - \tilde{\mathbf{w}}_t||^2\right] = \left(\frac{1}{c} + \delta_1\right) W^2 \tag{15}$$

**Proof:**

$$\mathbb{E}\left[||\mathbf{w}_t - \tilde{\mathbf{w}}_t||^2\right] = \mathbb{E}\left[||\mathbf{w}_t - \mathcal{U}_k(S(\mathbf{w}_t))||^2\right]$$

$$= \mathbb{E}\left[\sum_{i=1}^{k} |\mathbf{w}_t(i) - \tilde{\mathbf{w}}_t(i)|^2 + \sum_{i=k+1}^{d} (\mathbf{w}_t(i))^2\right]$$

$$= \mathbb{E}\left[\varepsilon^2 k ||\mathbf{w}_t||^2 + \sum_{i=k+1}^{d} (\mathbf{w}_t(i))^2\right]$$

$$= \mathbb{E}\left[\varepsilon^2 k \sum_{i=1}^{d}\left(\sum_{j=1}^{t} -\gamma\,\mathbf{g}_j(i)\right)^2 + \sum_{i=k+1}^{d} i^{-2p}\left(\sum_{j=1}^{t}||-\gamma\,\mathbf{g}_j||\right)^2\right]$$

$$= \mathbb{E}\left[\varepsilon^2 k \sum_{i=1}^{d} i^{-2p}\left(\sum_{j=1}^{t}\gamma\,||\mathbf{g}_j||\right)^2 + \sum_{i=k+1}^{d} i^{-2p}\left(\sum_{j=1}^{t}\gamma\,||\mathbf{g}_j||\right)^2\right]$$

$$= \left(\varepsilon^2 k + \frac{\sum_{i=k+1}^{d} i^{-2p}}{\sum_{i=1}^{d} i^{-2p}}\right)\mathbb{E}\left[\sum_{i=1}^{d} i^{-2p}\left(\sum_{j=1}^{t}\gamma\,||\mathbf{g}_j||\right)^2\right]$$

$$= (\varepsilon^2 k + \delta_1)\,\mathbb{E}\left[||\mathbf{w}_t||^2\right]$$

$$\leq (\varepsilon^2 k + \delta_1)\,W^2 = \left(\frac{1}{c} + \delta_1\right)W^2\,, \tag{16}$$

where, $\delta_1 (< 1)$ is given by the following expression:

$$\sum_{i=k+1}^{d} i^{-2p} = \delta_1 \sum_{i=1}^{d} i^{-2p},\quad \forall k \geq 1\,.$$

∎

Note that, the larger the sketch size gets; the number of coordinates that we can unsketch increases with higher accuracy ($\varepsilon$ decreases) and $\delta_1$ decreases as well. Also, $\delta_1$ is dependent on the the power law degree $p$ which is in turn dependent on the dataset.

### D.2 Lemma 2

**Lemma 2** *For a step size $\gamma \leq \frac{1}{4\,E\,(L+\mu)\,(1+2\,B^2(P_b+P_n))}$, we can bound the drift for any $e \in \{0,\dots,E-1\}$ as,*

$$\mathbb{E}\left[||\mathbf{w}_t^e - \mathbf{w}_t||^2\right] \leq 30\,E^2\,\gamma^2\left((1 + 2\,B^2\,(P_b + P_n))||\nabla f(\mathbf{w}_t)||^2 + b^2 + \sigma^2\right) \tag{17}$$

**Proof:** Now let us concentrate on the term $||\mathbf{w}_t^e - \mathbf{w}_t||^2$, we get:

$$
\begin{aligned}
\mathbb{E}\left[||\mathbf{w}_t^e - \mathbf{w}_t||^2\right] &= \mathbb{E}\left[||\mathbf{w}_t^{e-1} - \gamma\,\mathbf{g}_t^{e-1} - \mathbf{w}_t||^2\right] \\
&= \mathbb{E}\left[||\mathbf{w}_t^{e-1} - \mathbf{w}_t - \gamma\left(\mathbf{g}_t^{e-1} - \nabla f(\mathbf{w}_t^{e-1}) + \nabla f(\mathbf{w}_t^{e-1}) - \nabla f(\mathbf{w}_t) + \nabla f(\mathbf{w}_t)\right)||^2\right] \\
&\leq \left(1 + \frac{1}{2E-1}\right)\mathbb{E}\left[||\mathbf{w}_t^{e-1} - \mathbf{w}_t||^2\right] \\
&\qquad\qquad + 2\,E\,\gamma^2\,\mathbb{E}\left[||\nabla f(\mathbf{w}_t^{e-1}) - \mathbf{g}_t^{e-1} + \nabla f(\mathbf{w}_t^{e-1}) - \nabla f(\mathbf{w}_t) + \nabla f(\mathbf{w}_t)||^2\right] \\
&\leq \left(1 + \frac{1}{2E-1}\right)\mathbb{E}\left[||\mathbf{w}_t^{e-1} - \mathbf{w}_t||^2\right] + 6\,E\,\gamma^2\,\mathbb{E}\left[||\nabla f(\mathbf{w}_t^{e-1}) - \mathbf{g}_t^{e-1}||^2\right] \\
&\qquad\qquad + 6\,E\,\gamma^2\,\mathbb{E}\left[||\nabla f(\mathbf{w}_t^{e-1}) - \nabla f(\mathbf{w}_t)||^2\right] + 6\,E\,\gamma^2\,||\nabla f(\mathbf{w}_t)||^2 \\
&\leq \left(1 + \frac{1}{2E-1} + 6\,E\,(L+\mu)^2\,\gamma^2\right)\mathbb{E}\left[||\mathbf{w}_t^{e-1} - \mathbf{w}_t||^2\right] \\
&\qquad\qquad + 6\,E\,\gamma^2\left(\mathbb{E}\left[||\beta_t^{e-1} + \zeta_t^{e-1}||^2\right] + ||\nabla f(\mathbf{w}_t)||^2\right) \\
&\leq \left(1 + \frac{1}{2E-1} + 6\,E\,(L+\mu)^2\,\gamma^2\right)\mathbb{E}\left[||\mathbf{w}_t^{e-1} - \mathbf{w}_t||^2\right] \\
&\qquad\qquad + 6\,E\,\gamma^2\left(B^2\,(P_b + P_n)\,\mathbb{E}\left[||\nabla f(\mathbf{w}_t^{e-1})||^2\right] + b^2 + \sigma^2 + ||\nabla f(\mathbf{w}_t)||^2\right) \\
&\leq \left(1 + \frac{1}{2E-1} + 6\,E\,(L+\mu)^2\,\gamma^2\right)\mathbb{E}\left[||\mathbf{w}_t^{e-1} - \mathbf{w}_t||^2\right] \\
&\quad + 6\,E\,\gamma^2\left(B^2\,(P_b + P_n)\,\mathbb{E}\left[||\nabla f(\mathbf{w}_t^{e-1}) - \nabla f(\mathbf{w}_t) + \nabla f(\mathbf{w}_t)||^2\right] + b^2 + \sigma^2 + ||\nabla f(\mathbf{w}_t)||^2\right) \\
&\leq \left(1 + \frac{1}{2E-1} + 6\,E\,(L+\mu)^2\,\gamma^2\,(1 + 2\,B^2\,(P_b + P_n))\right)\mathbb{E}\left[||\mathbf{w}_t^{e-1} - \mathbf{w}_t||^2\right] \\
&\qquad\qquad + 6\,E\,\gamma^2\left((1 + 2\,B^2\,(P_b + P_n))||\nabla f(\mathbf{w}_t)||^2 + b^2 + \sigma^2\right).
\end{aligned}
$$

We assume $\gamma \leq \frac{1}{4\,E\,(L+\mu)\,(1+2\,B^2(P_b+P_n))}$ and using this in our analysis so far we get,

$$
\begin{aligned}
\mathbb{E}\left[||\mathbf{w}_t^e - \mathbf{w}_t||^2\right] &\leq \left(1 + \frac{1}{2E-1} + \frac{6}{16\,(1 + 2\,B^2\,(P_b + P_n))E}\right)\mathbb{E}\left[||\mathbf{w}_t^{e-1} - \mathbf{w}_t||^2\right] \\
&\qquad\qquad + 6\,E\,\gamma^2\left((1 + 2\,B^2\,(P_b + P_n))||\nabla f(\mathbf{w}_t)||^2 + b^2 + \sigma^2\right) \\
&\leq \left(1 + \frac{1}{2E-1} + \frac{1}{2\,E}\right)\mathbb{E}\left[||\mathbf{w}_t^{e-1} - \mathbf{w}_t||^2\right] \\
&\qquad\qquad + 6\,E\,\gamma^2\left((1 + 2\,B^2\,(P_b + P_n))||\nabla f(\mathbf{w}_t)||^2 + b^2 + \sigma^2\right) \\
&\leq \left(1 + \frac{1}{E-1}\right)\mathbb{E}\left[||\mathbf{w}_t^{e-1} - \mathbf{w}_t||^2\right] \\
&\qquad\qquad + 6\,E\,\gamma^2\left((1 + 2\,B^2\,(P_b + P_n))||\nabla f(\mathbf{w}_t)||^2 + b^2 + \sigma^2\right).
\end{aligned}
$$

Going recursively,

$$
\begin{aligned}
\mathbb{E}\left[||\mathbf{w}_t^e - \mathbf{w}_t||^2\right] &\leq \sum_{e=0}^{E-1}\left(1 + \frac{1}{E-1}\right)^e 6\,E\,\gamma^2\left((1 + 2\,B^2\,(P_b + P_n))||\nabla f(\mathbf{w}_t)||^2 + b^2 + \sigma^2\right) \\
&= (E-1)\left(\left(1 + \frac{1}{E-1}\right)^E - 1\right)6\,E\,\gamma^2\left((1 + 2\,B^2\,(P_b + P_n))||\nabla f(\mathbf{w}_t)||^2 + b^2 + \sigma^2\right) \\
&\leq 30\,E^2\,\gamma^2\left((1 + 2\,B^2\,(P_b + P_n))||\nabla f(\mathbf{w}_t)||^2 + b^2 + \sigma^2\right) \qquad\qquad (18)
\end{aligned}
$$

$\blacksquare$

The last inequality follows from the fact that $\left(1 + \frac{1}{E-1}\right)^E \leq 5$ for all $E > 1$. The proof of the above Lemma loosely follows the proof of Lemma 3 in Reddi et al. (2021). Let us now bound the second moment bounds

of computed stochastic gradient, bias and noise terms.

$$\mathbf{g}_t(\mathbf{w}_t) = \sum_{e=0}^{E-1} \mathbf{g}_t(\mathbf{w}_t^e). \tag{19}$$

Keeping the representation simple, we write $\mathbf{g}_t(\mathbf{w}_t^e) = \mathbf{g}_t^e$. Extending this representation, we can expand the computed gradient based on the general structure as, $\mathbf{g}_t^e = \nabla f(\mathbf{w}_t^e) + \beta_t^e + \zeta_t^e$.

$$\mathbb{E}\left[||\mathbf{g}_t||^2\right] = \mathbb{E}\left[\left|\left|\sum_{e=0}^{E-1} \nabla f(\mathbf{w}_t^e) + \beta_t^e + \zeta_t^e\right|\right|^2\right]$$

$$\leq E\left(\sum_e \left((2 + 2P_b\,B^2 + P_n\,B^2)||\nabla f(\mathbf{w}_t^e)||^2 + 2b^2 + \sigma^2\right)\right)$$

$$= (2 + 2P_b\,B^2 + P_n\,B^2)\,E\left(\sum_e ||\nabla f(\mathbf{w}_t^e)||^2\right) + 2\,E^2\,b^2 + E^2\,\sigma^2$$

$$= (2 + 2P_b\,B^2 + P_n\,B^2)\,E\left(\sum_e ||\nabla f(\mathbf{w}_t^e) - \nabla f(\mathbf{w}_t) + \nabla f(\mathbf{w}_t)||^2\right) + 2\,E^2\,b^2 + E^2\,\sigma^2$$

$$\leq 2\,(2 + 2P_b\,B^2 + P_n\,B^2)\,\underbrace{E\left(\sum_e ||\nabla f(\mathbf{w}_t^e) - \nabla f(\mathbf{w}_t)||^2\right)}_{\text{term 1}}$$

$$+ 2\,(2 + 2P_b\,B^2 + P_n\,B^2)\,E^2\,||\nabla f(\mathbf{w}_t)||^2 + 2\,E^2\,b^2 + E^2\,\sigma^2. \tag{20}$$

Focusing on bounding term 1 in the above equation, we get:

$$E\left(\sum_e \mathbb{E}\left[||\nabla f(\mathbf{w}_t^e) - \nabla f(\mathbf{w}_t)||^2\right]\right) \leq (L + \mu)^2\,E\sum_e \mathbb{E}[||\mathbf{w}_t^e - \mathbf{w}_t||^2]. \tag{21}$$

Using the result derived in Lemma 2 we get,

$$E\left(\sum_e \mathbb{E}\left[||\nabla f(\mathbf{w}_t^e) - \nabla f(\mathbf{w}_t)||^2\right]\right) \leq 30\,E^4\,(L + \mu)^2\,\gamma^2\,\left((1 + 2\,B^2\,(P_b + P_n))||\nabla f(\mathbf{w}_t)||^2 + b^2 + \sigma^2\right)$$

$$\leq \frac{2\,E^2}{(1 + 2\,B^2\,(P_b + P_n))^2}\,\left((1 + 2\,B^2\,(P_b + P_n))||\nabla f(\mathbf{w}_t)||^2 + b^2 + \sigma^2\right). \tag{22}$$

Now, using equation 22 in equation 20,

$$\mathbb{E}\left[||\mathbf{g}_t||^2\right] \leq 2\,E^2\,(2 + 2P_b\,B^2 + P_n\,B^2)\left(1 + \frac{2}{(1 + 2\,B^2\,(P_b + P_n))}\right)||\nabla f(\mathbf{w}_t)||^2$$

$$+ 2\left(1 + \frac{(2 + 2P_b\,B^2 + P_n\,B^2)}{(1 + 2\,B^2\,(P_b + P_n))^2}\right)E^2\,b^2 + 2\left(\frac{1}{2} + \frac{(2 + 2P_b\,B^2 + P_n\,B^2)}{(1 + 2\,B^2\,(P_b + P_n))^2}\right)E^2\,\sigma^2. \tag{23}$$

Similarly,

$$\|\beta_t\|^2 = \left\|\sum_{e=0}^{E-1} \nabla\beta_t^e\right\|^2$$

$$\leq E\left(\sum_e \left(P_b\,B^2\|\nabla f(\mathbf{w}_t^e)\|^2 + b^2\right)\right)$$

$$= P_b\,B^2\,E\left(\sum_e \|\nabla f(\mathbf{w}_t^e)\|^2\right) + E^2\,b^2$$

$$= P_b\,B^2\,E\left(\sum_e \|\nabla f(\mathbf{w}_t^e) - \nabla f(\mathbf{w}_t) + \nabla f(\mathbf{w}_t)\|^2\right) + E^2\,b^2$$

$$\leq 2P_b\,B^2\,E\left(\sum_e \|\nabla f(\mathbf{w}_t^e) - \nabla f(\mathbf{w}_t)\|^2\right) + 2P_b\,B^2\,E^2\,\|\nabla f(\mathbf{w}_t)\|^2 + E^2\,b^2\,.$$

Taking expectation on both sides and using the result derived in equation 22 we get,

$$\|\beta_t\|^2 \leq 2\,P_b\,B^2\,E^2\left(1 + \frac{2}{(1 + 2\,B^2\,(P_b + P_n))}\right)\|\nabla f(\mathbf{w}_t)\|^2 + \frac{2\,E^2}{(1 + 2\,B^2\,(P_b + P_n))^2}\left(\left(\frac{1}{2} + 2\,P_b\,B^2\right)b^2 + \sigma^2\right)\,. \tag{24}$$

Similarly the upper bound on the second moment of noise $\zeta_t$, we have

$$\mathbb{E}\left[\|\zeta_t\|^2\right] \leq 2\,P_n\,B^2\,E^2\left(1 + \frac{2}{(1 + 2\,B^2\,(P_b + P_n))}\right)\|\nabla f(\mathbf{w}_t)\|^2 + \frac{2\,E^2}{(1 + 2\,B^2\,(P_b + P_n))^2}\left(b^2 + \left(\frac{1}{2} + 2P_n\,B^2\right)\sigma^2\right)\,. \tag{25}$$

### D.3  Proof of Theorem 1

In this section, we begin by defining some quantities and notations. We define the quantity: $\tilde{\mathbf{w}}_{t+1} = \mathcal{U}_k(S(\mathbf{w}_{t+1}))$. Here, $\mathcal{U}_k(S(\cdot))$ represents the unsketching operation. The subscript $k$ denotes the number of top-$k$ coordinates extracted.

As defined in Assumption 1 of the paper, the application specific loss function is $L-$smooth. We denote this application specific loss function as $\ell(\cdot)$. For instance, for a binary classification task, the loss function can be log-loss. Now, our restructured loss function which is formulated by appending a proximal or regularizer term with the leading constant denoted as: $\mu$. This is given by:

$$f(\mathbf{w}, \mathbf{w}^{gb}) = \ell(\mathbf{w}) + \frac{\mu}{2}\left\|\mathbf{w} - \mathbf{w}^{gb}\right\|^2\,, \tag{26}$$

where, the iterate $\mathbf{w}^{gb}$ as the last aggregated model parameter vector that was broadcasted by the central server. To simplify, we reduce the notation of $f(\mathbf{w}, \mathbf{w}^{gb})$ to $f(\mathbf{w})$. Here, $\mathbf{w}$ is the current iterate at which the function is being evaluated. Appending such a proximal term preserves the smoothness of the function. Therefore, this new restructured loss function $f(\cdot)$ is $(L + \mu)-$smooth.

We assume that $\gamma \leq \frac{1}{2(L+\mu)}$. Given that $f(\cdot)$ is $(L+\mu)-$smooth,we have that:

$$
\begin{aligned}
\mathbb{E}_t[f(\tilde{\mathbf{w}}_{t+1})] &\leq f(\tilde{\mathbf{w}}_t) + \langle \nabla f(\tilde{\mathbf{w}}_t), \mathbb{E}_t[\tilde{\mathbf{w}}_{t+1} - \tilde{\mathbf{w}}_t] \rangle + \frac{(L+\mu)}{2} \mathbb{E}_t \left[ ||\tilde{\mathbf{w}}_{t+1} - \tilde{\mathbf{w}}_t||^2 \right] \\
&= f(\tilde{\mathbf{w}}_t) - \langle \nabla f(\tilde{\mathbf{w}}_t), \gamma \mathbb{E}_t[\mathbf{g}_t] \rangle + \frac{(L+\mu)}{2} \mathbb{E}_t \left[ ||\gamma \mathbf{g}_t||^2 \right] \\
&= f(\tilde{\mathbf{w}}_t) - \gamma \langle \nabla f(\mathbf{w}_t), \mathbb{E}_t[\mathbf{g}_t] \rangle + \langle \nabla f(\mathbf{w}_t) - \nabla f(\tilde{\mathbf{w}}_t), \gamma \mathbb{E}_t[\mathbf{g}_t] \rangle + \frac{(L+\mu)}{2} \gamma^2 \mathbb{E}_t \left[ ||\mathbf{g}_t||^2 \right] \\
&\overset{(a)}{\leq} f(\tilde{\mathbf{w}}_t) - \gamma \langle \nabla f(\mathbf{w}_t), \nabla f(\mathbf{w}_t) + \beta_t \rangle + \langle \nabla f(\mathbf{w}_t) - \nabla f(\tilde{\mathbf{w}}_t), \gamma \mathbb{E}_t[\mathbf{g}_t] \rangle \\
&\qquad + \gamma^2 (L+\mu) \left( ||\nabla f(\mathbf{w}_t) + \beta_t||^2 + \mathbb{E}_t \left[ ||\zeta_t||^2 \right] \right) \\
&\overset{(b)}{\leq} f(\tilde{\mathbf{w}}_t) + \frac{\gamma}{2} \left( -2 \langle \nabla f(\mathbf{w}_t), \nabla f(\mathbf{w}_t) + \beta_t \rangle + ||\nabla f(\mathbf{w}_t) + \beta_t||^2 \right) \\
&\qquad + \langle \nabla f(\mathbf{w}_t) - \nabla f(\tilde{\mathbf{w}}_t), \gamma \mathbb{E}_t[\mathbf{g}_t] \rangle + \gamma^2 (L+\mu) \left( \mathbb{E}_t \left[ ||\zeta_t||^2 \right] \right) \\
&= f(\tilde{\mathbf{w}}_t) + \frac{\gamma}{2} \left( -||\nabla f(\mathbf{w}_t)||^2 + ||\beta_t||^2 \right) + \langle \nabla f(\mathbf{w}_t) - \nabla f(\tilde{\mathbf{w}}_t), \gamma \mathbb{E}_t[\mathbf{g}_t] \rangle \\
&\qquad + \gamma^2 (L+\mu) \left( \mathbb{E}_t \left[ ||\zeta_t||^2 \right] \right),
\end{aligned}
\tag{27}
$$

where, inequality $(a)$ is a consequence of using Young's inequality. Inequality $(b)$ is a direct consequence of using the assumption $\gamma \leq \frac{1}{2(L+\mu)}$ from Lemma 2. To keep our analysis visually easy to follow we abbreviate the quantity $\frac{1}{1+2B^2(P_b+P_n)}$ as $H$.

Continuing on with our proof from Equation equation 27 and utilizing the second moment bounds from equation 23 , equation 24 and equation 25 we get:

$$
\mathbb{E}_t[f(\tilde{\mathbf{w}}_{t+1})] \leq f(\tilde{\mathbf{w}}_t) - \left( \frac{\gamma}{2} - \frac{\gamma\, P_b\,(1+2\,H)\,E^2\,B^2}{2} - 2\,P_n\,(L+\mu)\,(1+2\,H)\gamma^2\,E^2\,B^2 \right) \|\nabla f(\mathbf{w}_t)\|^2
$$

$$
+ 2\,E^2\,H^2 \left( \left( \frac{1}{2} + 2\,P_b\,B^2 \right)\gamma + \gamma^2\,(L+\mu) \right) b^2 + 2\,E^2\,H^2 \left( \gamma + \left( \frac{1}{2} + 2\,P_n\,B^2 \right)\gamma^2\,(L+\mu) \right)\sigma^2
$$

$$
+ \mathbb{E}_t[\langle (L+\mu)\,(\mathbf{w}_t - \tilde{\mathbf{w}}_t), \gamma\,\mathbf{g}_t \rangle]
$$

$$
\overset{(d)}{\leq} f(\tilde{\mathbf{w}}_t) - \left( \frac{\gamma}{2} - \frac{\gamma\, P_b\,(1+2\,H)\,E^2\,B^2}{2} - 2\,P_n\,(L+\mu)\,(1+2\,H)\gamma^2\,E^2\,B^2 \right) \|\nabla f(\mathbf{w}_t)\|^2
$$

$$
+ 2\,E^2\,H^2 \left( \left( \frac{1}{2} + 2\,P_b\,B^2 \right)\gamma + \gamma^2\,(L+\mu) \right) b^2 + 2\,E^2\,H^2 \left( \gamma + \left( \frac{1}{2} + 2P_n\,B^2 \right)\gamma^2\,(L+\mu) \right)\sigma^2
$$

$$
+ \frac{(L+\mu)^2}{2}\,\mathbb{E}_t \left[ \|\mathbf{w}_t - \tilde{\mathbf{w}}_t\|^2 \right] + \frac{\gamma^2}{2}\,\mathbb{E}_t \left[ \|\mathbf{g}_t\|^2 \right]
$$

$$
\leq f(\tilde{\mathbf{w}}_t) + \frac{(L+\mu)^2}{2}\,\boxed{\mathbb{E}_t \left[ \|\mathbf{w}_t - \tilde{\mathbf{w}}_t\|^2 \right]}
$$

$$
- \left( \frac{\gamma}{2} - \frac{\gamma\, P_b\,(1+2\,H)\,E^2\,B^2}{2} - 2\,P_n\,(L+\mu)\,(1+2\,H)\gamma^2\,E^2\,B^2 - \gamma^2\,E^2\,(2 + 2P_b\,B^2 + P_n\,B^2)\,(1+2\,H) \right) \|\nabla f(\mathbf{w}_t)\|^2
$$

$$
+ 2\,E^2\,H^2 \left( \left( \frac{1}{2} + 2\,P_b\,B^2 \right)\gamma + \gamma^2\,(L+\mu) + \left( \frac{1}{H^2} + (2 + 2P_b\,B^2 + P_n\,B^2) \right)\gamma^2 \right) b^2
$$

$$
+ 2\,E^2\,H^2 \left( \gamma + \left( \frac{1}{2} + 2P_n\,B^2 \right)\gamma^2\,(L+\mu) + \left( (2 + 2P_b\,B^2 + P_n\,B^2) + \frac{1}{2\,H^2} \right)\gamma^2 \right)\sigma^2 .
$$

$$(28)$$

Let us define the quantity:

$$
\rho(\gamma) = \frac{1 - P_b\,(1+2\,H)\,E^2\,B^2}{2} - 2\,P_n\,(L+\mu)\,(1+2\,H)\gamma\,E^2\,B^2 - \gamma\,E^2\,(2 + 2P_b\,B^2 + P_n\,B^2)\,(1+2\,H) .
$$

$$(29)$$

By averaging from 0 to $T$ on both sides and plugging the bound for residual term (highlighted in red in equation 28) by Lemma 1 the following holds with probability $1 - \delta$:

$$
\frac{1}{T+1} \sum_{t=0}^{T} \gamma\,\rho(\gamma)\,\|\nabla f(\mathbf{w}_t)\|^2 \leq \frac{|f(\mathbf{w}_0) - f(\mathbf{w}^*)|}{(T+1)} + \left( \frac{1}{c} + \delta_1 \right) \frac{(L+\mu)^2\,W^2}{2}
$$

$$
+ 2\,E^2\,H^2 \left( \left( \frac{1}{2} + 2P_b\,B^2 \right)\gamma + \gamma^2\,(L+\mu) + \left( \frac{1}{H^2} + (2 + 2P_b\,B^2 + P_n\,B^2) \right)\gamma^2 \right) b^2
$$

$$
+ 2\,E^2\,H^2 \left( \gamma + \left( \frac{1}{2} + 2P_n\,B^2 \right)\gamma^2\,(L+\mu) + \left( (2 + 2P_b\,B^2 + P_n\,B^2) + \frac{1}{2\,H^2} \right)\gamma^2 \right)\sigma^2 .
$$

Then using the fact that $H \leq 1$ and rearranging terms,

$$\frac{1}{T+1} \sum_{t=0}^{T} \rho(\gamma) \, ||\nabla f(\mathbf{w}_t)||^2 \leq \frac{|f(\mathbf{w}_0) - f(\mathbf{w}^*)|}{\gamma \, (T+1)} + \left( \frac{1}{c} + \delta_1 \right) \frac{(L+\mu)^2 \, W^2}{2}$$

$$+ \, 2 \, E^2 \left( \left( \frac{1}{2} + 2P_b \, B^2 \right) + \gamma \, (L+\mu) + \left( 1 + (2 + 2P_b \, B^2 + P_n \, B^2) \right) \gamma \right) b^2$$

$$+ \, 2 \, E^2 \left( 1 + \left( \frac{1}{2} + 2P_n \, B^2 \right) \gamma \, (L+\mu) + \left( (2 + 2P_b \, B^2 + P_n \, B^2) + \frac{1}{2} \right) \gamma \right) \sigma^2$$

$$\leq \frac{|f(\mathbf{w}_0) - f(\mathbf{w}^*)|}{\gamma \, (T+1)} + \left( \frac{1}{c} + \delta_1 \right) \frac{(L+\mu)^2 \, W^2}{2}$$

$$+ \, 2 \, E^2 \left( \left( \frac{1}{2} + 2P_b \, B^2 \right) + \frac{1}{4 \, E} + \frac{\left( 1 + (2 + 2P_b \, B^2 + P_n \, B^2) \right)}{4 \, (L+\mu) \, E} \right) b^2$$

$$+ \, 2 \, E^2 \left( 1 + \left( \frac{1}{2} + 2P_n \, B^2 \right) \frac{1}{4 \, E} + \left( (2 + 2P_b \, B^2 + P_n \, B^2) + \frac{1}{2} \right) \frac{1}{4 \, (L+\mu) \, E} \right) \sigma^2 . \tag{30}$$

### D.4 Alternate formulation of Theorem 1

Given under similar conditions of Theorem 1, where, the sketch size scales like $\mathcal{O}\left( c \, k \, \log \frac{d \, T}{\delta} \right)$, the learning rate satisfies the conditions: $\rho(\gamma) > 0$ and $\gamma \leq \frac{1}{2(L+\mu)}$, we can can cast Theorem 1 differently.

**Corollary 1** *Under assumptions 1,2,3,4 and 5, a fixed learning rate $\gamma$ then, for FPS after $T = \frac{\Delta}{\rho(\gamma) \, \epsilon}$ iterations the following statement holds with probability at least $1 - \delta$:*

$$\frac{1}{T+1} \sum_{t=0}^{T} ||\nabla f(\mathbf{w}_t)||^2 \leq \epsilon , \tag{31}$$

*where* $\Delta = \frac{|f(\mathbf{w}_0) - f(\mathbf{w}^*)|}{\gamma \, (T+1)} + \left( \frac{1}{c} + \delta_1 \right) \frac{(L+\mu)^2 \, W^2}{2} + \left( \left( 1 + 4 \, P_b \, B^2 \right) E^2 + \frac{E}{2} + \frac{\left( 1 + (2 + 2P_b \, B^2 + P_n \, B^2) \, E \right)}{2 \, (L+\mu)} \right) b^2$

$+ \left( 2 \, E^2 + \left( 1 + 4 \, P_n \, B^2 \right) \frac{E}{2} + \frac{\left( 2 \, (2 + 2P_b \, B^2 + P_n \, B^2) + 1 \right) E}{4 \, (L+\mu)} \right) \sigma^2 .$

Some interesting remarks that can be made based on this formulation:

- As the channel noise increases so does the bias and variance associated with it. As a consequence, the number of iterations it takes to converge increases.

- Recall the definition of $\rho(\gamma)$, it's magnitude decreases as the degree of statistical heterogeneity increases. We also see that the number of iterations to run FPS is inversely proportional to $\rho(\gamma)$. Therefore, as the degree of statistical heterogeneity increases, the number of iterations it takes to obtain desired result increases as well.

## E  Experimental details

### E.1  Setup

In this section, we provide more details on the experimental setup of our experiments. For the synthetic data set, we chose a mean-squared error loss function to minimize. The number of subcarriers allocated to each edge device is 256. The number of rows for count sketch data structure is 5 ,and the number of columns is given by the ceiling of ratio of number of subcarriers and number of rows. In this case, the ambient

dimension is 10000. The number of top-$k$ significant coordinates that we are extracting (unsketching) are 50.

For KDD12 real world dataset, we consider the number of subcarriers to be 1024. The number of rows for CS data structure are 5 and the number of columns are 204. The number of top-$k$ significant coordinates that we are extracting are 200. The ambient dimension of this dataset is 54,686,452.

For KDD10 real world dataset, we consider the number of subcarriers to be 4096. The number of rows for CS data structure are 5 and the number of columns are 820. The number of top-$k$ significant coordinates that we are extracting are 1000. The ambient dimension of this dataset is 20,216,830.

## E.2 Additional experiments

We present some more experimental results in this section. In Figure 5, we plot the performance of FPS, FetchSGD and BLCD for different data partitioning strategies mentioned in the main paper under noise-free channel conditions on KDD12 dataset. When the data is distributed in an IID manner (scenario 1), we see that FetchSGD performs slightly better than FPS. In scenario 2 where the data is highly heterogeneous, we see that FPS outperforms other competing bandlimited algorithms. In case of scenarios 3 and 4, we see that FPS matches the performance of FetchSGD.

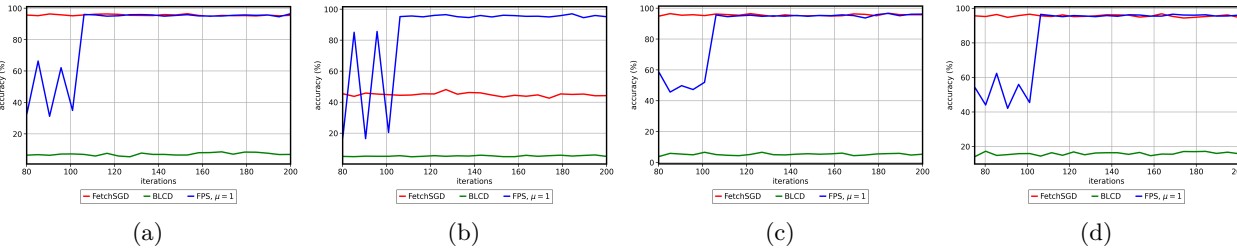

Figure 5: Plotting test accuracy for FPS, BLCD, FetchSGD on KDD12 dataset under noise-free channel conditions. The figures correspond to different data partitioning strategies: (a) Scenario 1 (b) Scenario 2 (c) Scenario 3 (d) Scenario 4.

In Figure 6, we plot the performance of FPS, FetchSGD and BLCD for different data partitioning strategies mentioned in the main paper under noise-free channel conditions on KDD10 dataset. Across all data partitioning scenarios we see that BLCD and FPS perform equally well and better than FetchSGD.

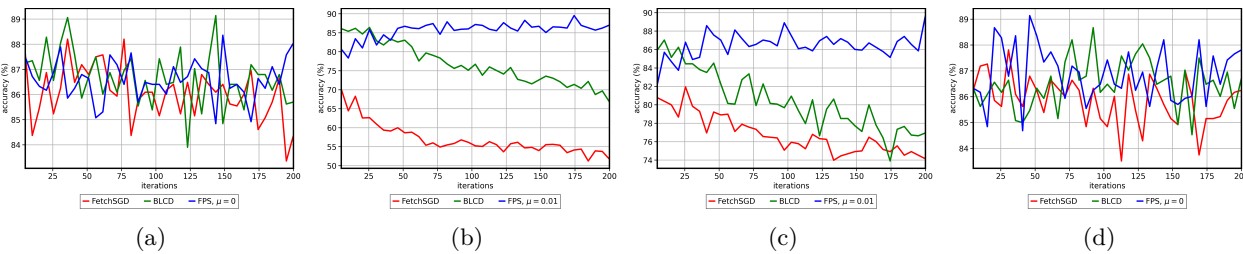

Figure 6: Plotting test accuracy for FPS, BLCD, FetchSGD on KDD10 dataset under noise-free channel conditions. The figures correspond to different data partitioning strategies: (a) Scenario 1 (b) Scenario 2 (c) Scenario 3 (d) Scenario 4.

## E.3 Choosing hyperparameters

There are two hyperparameters that we consider in the main paper that require further discussion. The first one is the choice of proximal parameter, $\mu$. A large value of $\mu$ will cause the future iterates to be close to the initialization iterate and a low value of $\mu$ may cause the model to diverge. Therefore, the value of proximal

parameter must be chosen carefully. In our experiments, we choose the best value of this proximal parameter from a set of values $\{0, 0.01, 0.1, 1\}$. For the two real-world data sets (KDD10 and KDD12) across different data partitioning strategies the best values of $\mu$ are 0.01 and 1 respectively. Note that picking the best value of $\mu$ right away is difficult due to varying statistical heterogeneity and different datasets. An interesting line of work could be finding the ideal choice of proximal parameter automatically. However, another interesting heuristic technique proposed in Li et al. (2020) adaptively tunes $\mu$. For instance, increase $\mu$ when the loss increases and vice versa. We have not examined the effects of such a heuristic in our experiments.

Another hyperparameter that we choose prior to the start of our experiments is number of local updates $E$ performed by each edge device. We choose a uniform $E = 5$ across all edge devices. Choosing a large value of E implies allowing large amounts of work done by edge devices and this can cause the model to diverge when the data is distributed in a non-IID manner. However, to mitigate this we have a proximal term which does not allow the local updates performed by the edge devices in this period to drift far away. However, the choice of an appropriate value of $E$ might be challenging problem in itself as it depends on device constraints and data distribution across all devices.

## F  Gradient compressibility

The idea that the computed stochastic gradients are compressible or approximately sparse is central to employ efficient compression techniques. In the main paper we formulate mathematically the approximately sparse behaviour of the computed gradients. This needs to be empirically validated as well. We consider the scenario where the data is distributed in an IID manner across devices. We run a federated learning algorithm where there is no bandwidth limitation i.e., high-dimensional gradient vectors are communicated. We consider noise-free channels and the updates are communicated to the central server at every iteration. The loss function has no proximal term appended to it. This naive setup will help us understand the true behaviour of computed stochastic gradients. We run this vanilla FL algorithm for 200 iterations and at the end of it we report $\sim 90\%$ accuracy on both real world datasets (KDD10 and KDD12).

The number of features in the datasets KDD10 and KDD12 are 20,216,830 and 54,686,452 respectively. In Figures 7(a) and 8(a), we plot the absolute value of gradient coordinates computed at a particular edge device for the datasets KDD10 and KDD12 respectively. This plot is captured across three time instants, at iteration 25, 75 and 150. We see that in both figures, the absolute value of coordinates of the local gradient vector sorted in decreasing order are approximately sparse or follow a power law distribution. Similarly in

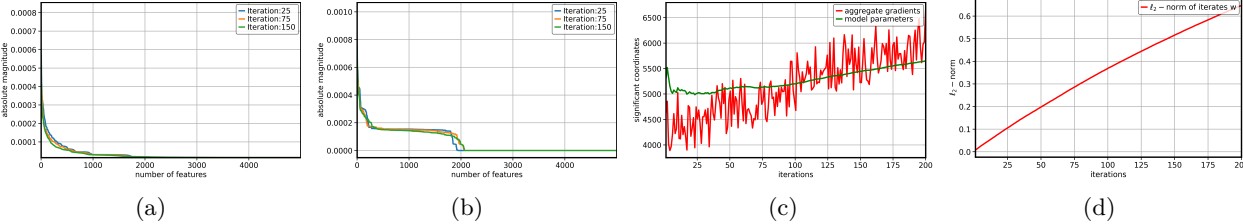

(a)         (b)         (c)         (d)

Figure 7: KDD 10 Dataset (a) sorted stochastic gradient at a single edge device (b) sorted aggregated stochastic gradient at the central server (c) significant coordinates of aggregated gradient vector and iterates at the central server (d) $\ell_2-$ norm of iterates.

Figures 7(b) and 8(b) we plot the absolute value of coordinates of the aggregated gradient vector received at the central server sorted in decreasing order. This plot is captured across three time instants, at iteration 25, 75 and 150. We observe a similar approximately sparse or power law behaviour for aggregated gradient vectors. If we approximate the number of significant coordinates in computed gradient vectors just by visual inspection of the plots, it is less than 3000. This is far less than the ambient dimension of the datasets we are operating on.

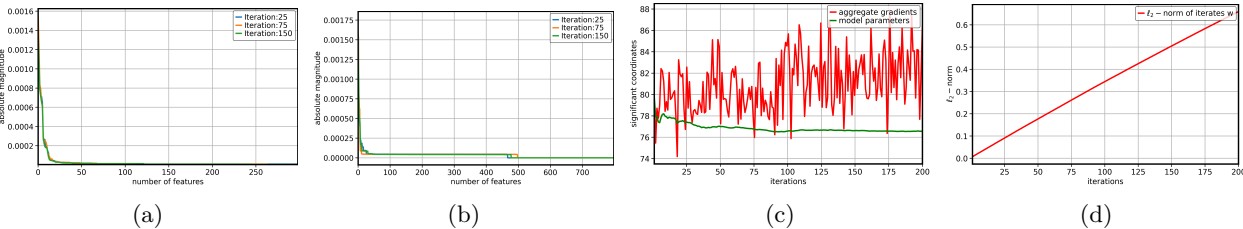

(a)            (b)            (c)            (d)

Figure 8: KDD 12 Dataset (a) sorted stochastic gradient at a single edge device (b) sorted aggregated stochastic gradient at the central server (c) significant coordinates of aggregated gradient vector and iterates at the central server (d) $\ell_2-$ norm of iterates.

However, a stronger notion of significant coordinates needs to be used. To this extent we use an alternative measure called *soft* sparsity defined in Lopes (2016):

$$sp(\mathbf{x}) = \frac{||\mathbf{x}||_1^2}{||\mathbf{x}||_2^2} \tag{32}$$

Soft-sparsity represents the number of significant coordinates in a vector. Let $\mathbf{g}$ and $\mathbf{w}$ denote the aggregated gradient and the model parameter vector respectively. For KDD10 dataset, the number of significant coordinates for the aggregated gradient vector $sp(\mathbf{g})$ and the model parameter vector $sp(\mathbf{w})$ are $\sim 5000$, which is much smaller than the ambient dimension. Similarly, for KDD12 dataset, the the number of significant coordinates for the aggregated gradient vector $sp(\mathbf{g})$ are $\sim 85$ and the model parameter vector $sp(\mathbf{w})$ are $\sim 75$. This can be seen in Figures 7(c) and 8(c).

Additionally, we show that the $\ell_2-$norm of the iterates at every iteration received at the central server does not explode and can be uniformly bounded above by a constant. This can be seen in Figures 7(d) and 8(d) for datasets KDD10 and KDD12 respectively.

## G    Dealing with bias

The vanilla stochastic gradient descent has been well studied in presence of unbiased gradient updates Bottou et al. (2018). Recently, biased gradient updates have been considered in SGD, for instance, in large-scale machine learning systems techniques sparsification, quantization have been used to mitigate the issue of communication bottleneck. Such compression techniques produce biased gradient updates. There is a growing line of work on how different error accumulation and feedback schemes can mitigate the issue of bias and speed up convergence of SGD and distributed learning algorithms Karimireddy et al. (2019); Stich et al. (2018). More recent work on error feedback can be found in Gorbunov et al. (2020); Qian et al. (2021).While this is not the focus of our paper, we are more interested in understanding how bias plays a role in theoretical convergence analysis of SGD. To this extent, we turn towards the body of literature that has dealt with modeling bias into the stochastic gradient structure. Our main motivation to have a more general stochastic gradient structure and mild conditions on bias and noise comes from the work in Ajalloeian & Stich (2020a). Additional works that have considered similar assumptions are Stich (2019); Hu et al. (2021); Bottou (2010) We believe utilizing the assumptions from this line of work into distributed optimization literature (for our paper, FL to be precise) can help us analyze algorithms on a broader scale.

