# OpenReview forum: "Communication Efficient Federated Learning over Wireless Channels using Robust Count Sketches"
_TMLR — Rejected by TMLR_

### Review · Reviewer_quXH · 2023-06-05

**Summary Of Contributions:**

In this paper, the authors proposed a new federated learning algorithm called Federated Proximal Sketching (FPS) to tackle the issues of communication bottleneck, noisy wireless environments, and data heterogeneity in FL. Specifically, the authors exploit count sketch to compress the model parameters and reconstruct the loss function with an additional proximal term. The convergence analysis of the proposed algorithm is provided, and the experimental results further validate the effectiveness of the proposed algorithm.

**Audience:**

Yes

**Broader Impact Concerns:**

N.A.

**Claims And Evidence:**

Yes

**Requested Changes:**

## Critical Changes (Necessary for Acceptance)
It is better to provide some additional explanations on the motivation for this work.

The authors need to determine whether this work is based on distributed SGD or FL. If it is based FL, the convergence analysis needs to be revised to accommodate multiple local epochs.

For the experimental setup, FPS perform global aggregation after every 5 local epochs. However, for each global iteration, BLCD and FetchSGD perform only one local epoch in the client side (they are both distributed SGD methods). I don't think that's a fair setting. As mentioned above, one of the central questions the authors need to clarify is whether FPS is a distributed SGD method or a federated learning algorithm.

It would be nice to include more FL schemes as experimental baseline.

I am curious about the result in Table 1. It seemed that FedProx do not perform well under noisy channel conditions, which probably indicates that adding proximal term in the loss function can not overcome the noisy environments. However, FPS use the exactly same way and got a success. I think more explanation needs to be added to illustrate the effectiveness of FPS.



## Recommended Changes

 Since the Count Sketch and the loss function with proximal term is independent of each other, it is interesting to perform ablation study to verify their respective effects.

Typos:

In Definition 2, $||E_{\xi_m}[\nabla f_m(w;\xi_m)]||^2$  should be $E_{\xi_m}[||\nabla f_m(w;\xi_m)||^2]$.

In Lemma 1, $E[||w_t-\tilde{w_t}||]$ should be $E[||w_t-\tilde{w_t}||^2]$.

In the second line of proof of Lemma 1, $\sum_{i=1}^k|w_t(i)-\tilde{w}_t(i)|$ should be $\sum_{i=1}^k|w_t(i)-\tilde w_t(i)|^2$.

**Strengths And Weaknesses:**

## Strengths

The considered problem is well-motivated and interesting.

The paper is well-written and easy to follow.

The proposed algorithm has convergence guarantee. The theoretical analysis is solid and in-depth.

Numerical result shows that the proposed algorithm can outperform the baseline especially under noisy channel conditions.



## Weaknesses

This work aims to tackle three challenges: a) large communication cost, b) noisy network environment, and c) data heterogeneity among clients. The author uses the Count Sketch to compress model parameters to deal with a). For b) and c), the author adopted the Loss function in FedProx. In my opinion, these two designs are independent of each other, which is just a simple A+B pattern. There is a lack of connection between them.

The framework described in Section 3.1 is distributed SGD, not Federated learning. The difference between them is the number of local iterations on the client side. Distributed SGD only computes a minibatch gradient on the client side, whereas FL performs local SGD of multiple epochs.

In Algorithm 1, line 6, the author mentions that E local iterations are performed on the client side between each global communication round. However, in the Convergence Analysis section, it seems that E is set to 1. This is not a reasonable assumption. In fact, under data heterogeneous scenarios, when E increases, the drift of the local model from the collaborative convergence will also increase (see Lemma 3 in [1]). Consequently, I think Hypothesis 3 may be incorrect. As E increases, the upper bound of the biased estimation error must increase.

 [1] Reddi, S., Charles, Z., Zaheer, M., Garrett, Z., Rush, K., Konečný, J., ... & McMahan, H. B. (2020). Adaptive federated optimization. arXiv preprint arXiv:2003.00295.

---

> ### Author Response · Authors · 2023-06-30
> **Response to reviewers**
>
> We would like to thank the reviewers for providing insightful and constructive comments. We have tried our best to address all the comments individually. We note that since some of the comments overlapped / were an extension of others, we have clubbed them into one `meta-point'  and provided a response.
>
> We have fixed the grammatical errors and minor mathematical errors pointed out by reviewers (quXH, uAW2). We have also tired our best to move the figures around such that to avoid flipping multiple pages.
>
> ## Clarifying the motivation of our work.
> We outline the motivations in the following:
>
> One main motivation is to design an algorithm that tackles some of the challenges in federated learning in more realistic settings and  analyze its convergence. The three major challenges listed have been  studied either independently or in pairs. Studying these challenges together, and understanding the interplay between various parameters, while still providing  sensible upper bounds, was the main objective of this paper.
>
> Further, we also strive to   analyze the proposed FL algorithm under the  general  conditions, such as the general stochastic gradient structure. The assumptions  in our work serve to analyze not only our algorithm but also other FL algorithms on a broader scale.
>
> The design of our algorithm was mainly divided into two segments:
> a) Choosing the CS data structure to tackle the bandlimited issue. The motivation for this is discussed in the next question.
> b) The proximal term is actually  a very popular technique in the optimization literature, e.g., it has been used to make NNs more robust to noise and promote generalizability [1,2] . Moreover, by slightly modifying the proximal term (as shown in [3]), it was demonstrated that it can avoid diverging when statistical heterogeneity is present across devices. Therefore, motivated by the fact that the proximal term can tackle two issues, we utilize it in our algorithm design.
>
> [1]C. M. Bishop, “Regularization and complexity control in feed- forward networks,” 1995
>
> [2]C. M. Bishop, “Training with noise is equivalent to tikhonov regularization,”  1995.
>
> [3]Tian Li, Anit Kumar Sahu, Manzil Zaheer, Maziar Sanjabi, Ameet Talwalkar, and Virginia Smith. Federated optimization in heterogeneous networks. In Proceedings of Machine Learning and Systems, 2020
>
> ## Choice of baselines
>
> Since we use sketching as our compression technique, FetchSGD is one of the state-of-the-art FL algorithms that employs the CS data structure to communicate compressed gradient updates to the central server. Moreover, in the original paper where FetchSGD was proposed, the authors conducted experiments to tackle non-iid data distribution across edge devices as well. This makes it a fair baseline for  the comparison of our algorithm, FPS. BLCD was chosen as another baseline as it uses sparsification as a compression operator. Sparsification produces biased gradient updates, and recent work has shown that convergence can still be achieved by employing an error accumulation feedback. Additionally, BLCD is perfectly bandlimited, not requiring any additional rounds of communication, unlike top-k sparsification. All of the above reasons make it a suitable FL bandlimited baseline. Since our algorithm is motivated by FedProx (in its usage of the proximal term), we include it in our comparison as a baseline.

---

> > ### Author Response · Authors · 2023-06-30
> > **Response to reviewers continued.**
> >
> > ## Choice of datasets like KDD10 and 12.
> > While there is a plethora of popular datasets like MNIST, CIFAR, and their variants that we could use for empirical analysis, our work primarily focuses on the theoretical analysis of our proposed algorithm.
> > We chose binary classification datasets like KDD10 and KDD12 because they are high-dimensional, and gradient update communication can potentially become a bottleneck. It is worth noting that large-scale datasets like KDD10 and KDD12 are very common in modern applications such as predicting click-through rates and genomics. Some of the other high dimensional datasets that caught our attention were:
> > Webspam-Trigram - https://www.csie.ntu.edu.tw/~cjlin/libsvmtools/datasets/binary.html#webspam,
> > DNA Metagenomics - http://projects.cbio.mines-paristech.fr/largescalemetagenomics/,
> > Criteo 1TB - https://www.csie.ntu.edu.tw/~cjlin/libsvmtools/datasets/binary.html#criteo_tb,
> > Splice-Site 3.2TB - https://www.csie.ntu.edu.tw/~cjlin/libsvmtools/datasets/binary.html#splice-site.
> >
> > We would also like to add our experiments are not extensive but serve as a representation of theoretical findings. The main aim of the paper is to have a theoretically novel way of analyzing FL algorithms which captures some of the challenges.
> >
> > ## Effect of local epochs E in convergence analysis
> >
> >  Per your suggestion, we have also extended our theoretical analysis based on the provided review to include the effect of the number of local epochs on our convergence result. The details are summarized in  Lemma 2 and in the revised proof of Theorem 1. The remarks after Theorem  1 in the paper have been updated to provide more intuitive insight into the selection of hyperparameters ($\mu, E, \gamma$).

---

> > > ### Author Response · Authors · 2023-06-30
> > > **Response to reviewers continued.**
> > >
> > > ## Does our work fall under D-SGD or FL?
> > > Federated learning focuses on training models on decentralized data while preserving data privacy by training models locally on devices and aggregating updates. Our work falls in the FL domain, as the data is decentralized and privacy of it is important. This is inline with the definition discussed in [1] on Page 4. While distributed SGD assumes a centralized dataset divided among workers and aims to parallelize the computation across multiple machines for faster training. There is a fine line between D-SGD and FL, the distinguishing difference being that the edge devices in FL are capable of accumulating their local dataset which is stored locally and not exchanged or transferred.
> > >
> > > Moreover, it is not always the case to perform multiple local epochs  at each client between two global aggregation time steps for a FL algorithm. The number of local epochs can be a design choice: For instance, in our case we have local epochs before global aggregation due to heterogeneity of data distribution across clients. Another reason for choosing multiple local epochs can be system specific - if there are stragglers (slow devices), this can cause latency issues and the aggregating after a few local iterations can lead to better convergence results, or due to bandwidth limitations where only a few communication rounds can be allowed.  Here are few FL algorithms references which perform global aggregation at every time step [2,3,4].
> > >
> > > [1]  Kairouz, H. Brendan McMahan, Brendan Avent, Aurélien Bellet, Mehdi Bennis, Arjun Nitin Bhagoji, Advances and open problems in federated learning.
> > >  Found. Trends Mach. Learn.2021
> > > [2] Rabbani et. al. 2021 Comfetch- Federated Learning of Large Networks on Memory-Constrained Clients via Sketching
> > > [3]Xizixiang Wei Cong Shen. 2021. Federated Learning over Noisy Channels:Convergence Analysis and Design Examples
> > > [4] Rothchild et.al. 2020. FetchSGD- Communication-Efficient Federated Learning with Sketching
> > >
> > > ## Performance of FedProx under non-iid and noisy data
> > >
> > > Upon analyzing the experimental studies conducted in the FedProx paper [5], we noticed that the chosen datasets are not particularly high dimensional. This is in contrast to the large-scale high dimensional datasets considered in our work. Specifically, the datasets we consider have only a fraction of important features, resulting in approximately sparse gradient updates that align with Assumption 4 as defined in our paper.
> > >
> > > One possibility is that FedProx, by design, is not  bandlimited. As a result running simulation on KDD10 or KDD12, a significant number of coordinates may be corrupted (due to approximately sparse gradient updates/ only a few model parameters are heavy hitters), especially when the value of the proximal constant $\mu$ is chosen to be too small. Choosing a higher value for the proximal parameter has the potential to address the convergence issue in the presence of channel noise when using FedProx. Another possible solution is to use a longer training time.
> > >
> > > [5] Tian Li, Anit Kumar Sahu, Manzil Zaheer, Maziar Sanjabi, Ameet Talwalkar, and Virginia Smith. Federated optimization in heterogeneous networks. In Proceedings of Machine Learning and Systems, 2020

---

### Review · Reviewer_uAW2 · 2023-06-12

**Summary Of Contributions:**

The authors propose an algorithm that modifies the FetchSGD algorithm in (Rothchild et al) by (1) adding a regularization term into the drift between local and global updates (eq. 6) when computing the gradient and (2) maintaining a sketch of the current model parameters and updating it by sketching in the gradient update (- \eta \nabla). They provide a proof that captures the impact of bias in the gradient estimate. The authors evaluate on two datasets, KDD10 and KDD12.

**Audience:**

Yes

**Claims And Evidence:**

Yes

**Requested Changes:**

# Major:

Comprehensive evaluation: If you are making a claim that your method outperforms prior work empirically, you should evaluate both methods in at least the setting that prior work engages with. In this case, to substantiate this claim you would need to evaluate your method when optimizing ResNets on CIFAR or FEMNIST.

Code release: I tried experimenting with the synthetic dataset mentioned in 6.1 with the simple global top-k method but I was not able to fully grasp the implementation. Providing code for your proposed method would aid in reproducibility of the empirical evaluation, and I think this is important because this work is primarily empirical.

# Minor:

There are some minor typos (e.g., greatre than zero), please make sure that a cleaning pass is done for grammar/spelling.

Algorithm 1 should really be on the same page as Section 4. It's a bit confusing to keep referring back and forth across 2 pages. I ended up just folding Page 7 so that I could artificially have them on the same page.

Similarly, the results are not close to the text that analyzes them. Again, flipping back and forth is kind of annoying. It would be good if instead of repeating the line `the figures correspond to ...' there can be some text in the caption that gives a preliminary analysis of the figures and tables.



**Strengths And Weaknesses:**

# Strengths:

## Theory

The authors clearly state all 5 assumptions needed to provide their main Theorem 1 and I see no issue with any of these assumptions.
Furthermore the theoretical guarantees in (Rothchild et al) hold only for the non-iid setting where the stochastic gradient is unbiased but Theorem 1 allows for the estimate to be biased and the proposed method will degrade in the presence of bias, whereas the degradation of FetchSGD in the presence of bias is somewhat unknown.

## Empirical

According to the empirical evaluation, the proposed method outperforms FetchSGD in Scenario 2, where the data is highly heterogeneous. This is interesting because the evaluation in (Rothchild et al) follows a Dirichlet distribution with \alpha=0.0 (that is, each device holds only elements from a single class). But the class-imbalanced evaluation in (Rothchild et al) is only done on benchmark deep learning datasets, that is, CIFAR10, CIFAR100, FEMNIST, and the heterogeneity between image classes may still be mild compared to other types of statistical heterogeneity. In particular, it seems that FetchSGD falls apart completely when the noise is greater than 0 (Table 1) but the proposed method somehow manages to do even better! (Scenario 4)

# Weaknesses

## Novelty

I can't say that the extensions on top of FetchSGD are very novel. The analysis is more novel, but this is tempered by the fact that Theorem 1 does not seem to recover the convergence rate of SGD. I'm not sure what the significance of (2) is beyond eliminating the need to maintain the error accumulation vector server-side. Just adding a regularization term is not really novel, there are many prior works in FL that add a regularization term into the model drift. Indeed, there are papers that don't even mention this but just include it in the code because it is easy to implement.

## Theory

The introduction of bias into the stochastic gradient as the result of over-the-air communication via subcarriers is warranted, but the paper does not make it clear to what extent prior work has engaged with bias. For instance, it's not clear to me that prior algorithms have *no* guarantees in the presence of bias, rather that something similar to the analysis of Theorem 1 (cannot converge in the presence of too much bias) is probably true but underanalyzed. For example, (Khirirat et al) propose a framework for better studying the impact of bias in the gradient estimate under Top-k and cite a line of prior work that engages with the issue of bias.

## Empirical

I'm pretty unsure about the hyperparameter selection strategies. It is known that FetchSGD is reliant on the correct choice of hyperparameters, and the proposed method is also reliant on the correct choice of hyperparameters. The authors mention how they tune the regularization parameter \mu, and how they choose the local epochs E, k, and count sketch size. However the issue is that the authors tune the hyperparameters for their method but not the baselines. This is by no means a unique fault of the paper -it's more of a universal fault of all deep learning methods- but I must bring it up nevertheless.

I'm not sure about the datasets that are used for evaluation. Prior work in FL typically evaluates on benchmark CV/NLP datasets, or datasets from the LEAF benchmark, or ideally a mixture of these. The authors make a strong claim that their method is more stable, accurate and efficient than prior work with extensive experiments but I really can't say that evaluating on 2 datasets can be considered extensive.

Scenario 2 in particular seems underexplored. If KDD12 is a binary classification, then it doesn't seem like the experiments on this dataset really engage with class imbalance much compared to doing experiments on CIFAR100 where the imbalance is 1 vs 99 rather than just 1 vs 1.

Citations;
(Rothchild et al) - https://arxiv.org/pdf/2007.07682.pdf
(Khirirat et al) - https://arxiv.org/abs/2305.18929

---

> ### Author Response · Authors · 2023-06-30
> **Response**
>
> We thank the reviewer for their comments and questions.
>
>
> To answer the question about our empirical evaluation, we would like the reviewer to refer to our answer for sections below titled 'Motivation of our work', 'Choice of baselines' and 'Choice of datasets'.
>
> ## Clarifying the engagement of prior work with bias
>
> We appreciate the reviewer bringing this to our attention. A central question considered in this paper is to quantify the impact of bias in our work. We have added an additional paragraph explaining our motivation and list prior publications which study bias in Appendix G.
>
> ## Source Code
>
> The source code can be found at : https://anonymous.4open.science/r/FL-FPS-BF3A/
>
> ## Class imbalance is  1 vs 99 (in CIFAR100 dataset) rather than just 1 vs 1 like in current work.
>
> We appreciate the reviewer for bringing this to our attention. Indeed, a severe class imbalance where most workers have access to only one class out of 100 would lead to convergence issues. In such a scenario, the value of the dissimilarity constant $B$ would be very high. Consequently, we have elaborated on how increased data heterogeneity affects our convergence result in the remarks of our theorem. It would be an interesting experiment to observe the performance of our proposed algorithm in a 1 vs 99 class imbalance setting. However, for this paper, our focus was on demonstrating our novel and more general theoretical framework, as well as providing meaningful insights into the interplay between different variables. The numerical experiments, while not comprehensive, serve to complement our findings.

---

### Review · Reviewer_wYWV · 2023-06-20

**Summary Of Contributions:**

This paper studies the communication problem in federated learning. Specifically, the paper aims to address communication efficiency and non-iid issues.
The paper repurposed the Count Sketch data structure for gradient compression, with a provable bound on the information loss.
Furthermore, an L2 regularization is added to the loss to alleviate the heterogeneous error accumulated from consecutive local updates.
Theoretically, the authors provide a complete convergence analysis of the resulting algorithm.
Empirical evaluation is conducted on KDD 10/12 and synthetic datasets, and the proposed algorithm consistently outperforms the two selected baselines: FetchSGD and BLCD under bandlimited setting, as well as FedProx on handling non-iid cases.

**Audience:**

Yes

**Broader Impact Concerns:**

Not to my knowledge

**Claims And Evidence:**

Yes

**Requested Changes:**

1. If plausible, including more baseline algorithms for a comprehensive comparison with prior work.

**Strengths And Weaknesses:**

[Strength]
1. To the best knowledge, the use of Count Sketch for gradient compression is novel and has not been explored in prior works.
2. Empirical results show that the proposed algorithm consistently outperforms selected baselines, often by a substantial margin.
3. The author also provides a complete convergence analysis of the proposed algorithm.

[Weakness]

1. Gradient compression is one of the core techniques in FedML and many prior methods have been explored. From the current manual script, it is not clear to me intuitively why Count Sketch is superior to prior methods.
2. Following 1, the main experiments only benchmarked two band-limited baselines, FetchSGD and BLCD. Can the authors elaborate more on the reason for picking those algorithms out of a large body of gradient compression-fed algorithms?
3. The main real dataset of use is the KDD dataset. Can the author also elaborate on its choice and why other commonly used datasets such as MNIST and CIFAR variants are excluded?

---

> ### Author Response · Authors · 2023-06-30
> **Response**
>
> We thank the reviewer for bringing up some good questions.
>
> ## Why use count-sketch ?
>  We chose Count Sketch as a compression operator because of its theoretical guarantees on the recovery of heavy hitter coordinates. Using Theorem 2 (Charikar et al., 2002) in Appendix C, we demonstrate how the CS data structure helps us visualize the interplay between the number of subcarriers (CS data structure size) available and the convergence, to be more specific the residual error term after unsketching (term 2 on the right hand side of Equation (14)). Moreover, the CS data structure allows for an efficient representation (top-k heavy hitters) of model parameters at all time steps across all edge devices. This is due to the linearity aggregation property of CS, i.e.,  $S(w_{t-1}) + S(-\gamma g_t) = S(w_{t-1} - \gamma g_t) = S(w_t)$. This linearity aggregation property of CS data structures makes it easy to implement with over-the-air communication protocol . $S(w^1_t) + S(w^2_t) + \dots = S(w^1_t + w^2_t + \dots)$.
>
> We would also like to justify Assumption 4, as it provides a wide range of gradient vector types, from heavy-tailed to exponentially decaying (approximately sparse). By utilizing the theoretical guarantees of the CS data structure and Assumption 4, we were able to obtain the trade-off between the CS data structure size and the sparsity of the gradients. This analysis of residual error by utilizing a compression technique such as CS is novel.
>
>
> The answers to the other two questions are provided below under headings 'Choice of baseline' and 'Choice of datasets'.

---

### Decision · Action_Editors · 2023-08-07

**Recommendation:** Reject

**Comment:**

The opinions of the reviewers are divided, with a majority of the reviewers proposing a reject decision.

The experimental results could not fully convince the reviewers. However, they recognize that the theoretical findings make up the main part of the manuscript and that these results should form the basis for the decision and not the experiments alone.

The theoretical results,
- were found to be correct, derived under assumptions that are clearly stated (+),
- however, the relationship between the parameters (such as how $P_b, b$ depend on $c, k$) is not clear/discussed, so that the precise convergence properties of the proposed FPS remain a bit too vague (-).

Furthermore, the reviewers would have enjoyed a discussion of cases when FPS can be advantageous over other methods (considering the bias and the constraints on the data-heterogeneity in Theorem 1) or when other SOTA methods should could be preferred. (-)

The reviewers believe that the TMLR community's interest in the current results is therefore limited, but these concerns could potentially be addressed in a major revision at a later time.



**Audience:**

Communication efficient FL learning algorithms are of interest to parts of the TMLR audience, but the reviewers argue that the interest in the findings presented in this work might be too limited (see below).

**Claims And Evidence:**

This paper extends the FedProx framework with uplink compression based on a count sketch data structure. A convergence analysis is given under the assumption that the data transmission is corrupted with noise, and considering the bias introduced by the sketching of the parameters.

The reviewers commented that the empirical evaluation is a bit limited - sufficient to illustrate the main features of the algorithm, but not comprehensive enough to demonstrate the benefit of the proposed scheme across diverse tasks and against SOTA compression techniques.

The method is analyzed under a set of assumptions that are clearly stated. The first version of the submitted manuscript focused on the distributed optimization case without local steps ($E=1$) but during the review process the results have been extended to the federated setting with local steps ($E>1$). The theoretical results do not corroborate the claim that the proposed method can handle high levels of data heterogeneity across edge devices ($P_b B^2 E^2 \leq 1$).





**Resubmission Of Major Revision:**

The authors may consider submitting a major revision at a later time.